# Chronic inflammation degrades CD4 T cell immunity to prior vaccines in treated HIV infection

Melissa Kießling[1], John J. Cole[2], Sabrina Kübel [1], Paulina Klein[1], Klaus Korn [1], Amy R. Henry[3], Farida Laboune[3], Slim Fourati [4], Ellen Harrer[5], Thomas Harrer [5], Daniel C. Douek [3], Klaus Überla [1] & Krystelle Nganou-Makamdop [1,6] ✉

To date, our understanding of how HIV infection impacts vaccine-induced cellular immunity is limited. Here, we investigate inflammation, immune activation and antigen-specific T cell responses in HIV-uninfected and antiretroviral-treated HIV-infected people. Our findings highlight lower recall responses of antigen-specific CD4 T cells that correlate with high plasma cytokines levels, T cell hyperactivation and an altered composition of the T subsets enriched with more differentiated cells in the HIV-infected group. Transcriptomic analysis reveals that antigen-specific CD4 T cells of the HIV-infected group have a reduced expression of gene sets previously reported to correlate with vaccine-induced pathogen-specific protective immunity and further identifies a consistent impairment of the IFNα and IFNγ response pathways as mechanism for the functional loss of recall CD4 T cell responses in antiretroviral-treated people. Lastly, in vitro treatment with drugs that reduce inflammation results in higher memory CD4 T cell IFNγ responses. Together, our findings suggest that vaccine-induced cellular immunity may benefit from strategies to counteract inflammation in HIV infection.

The effect of HIV infection on vaccine-induced humoral immunity has long been acknowledged. Primarily, studies assessing the efficacy of licensed vaccines have demonstrated that untreated HIV infection lowers the percentage of person developing Hepatitis B protective titer[1] and reduces the frequencies of persons responding to measles and tetanus toxoid vaccines[2,3]. Among antiretroviral (ART)-treated adults, various studies have shown low seroconversion rates and more rapidly waning concentrations of antibodies against hepatitis A, hepatitis B, measles, tetanus toxoid and rubella[3–6]. While antibody responses are a common outcome measure in vaccine studies, most

vaccines induce both antibody and T cell responses. In fact, protective immunity may rely on T cell responses in absence of antibodies[7,8]. With respect to T cell responses, it was shown that people with HIV under ART have lower vaccinia-virus-specific IFNγ and TNF responses of CD4 T cells despite similar CD8 T cell responses[9] and lower proliferative capacity of tetanus toxoid-, mumps- and influenza-specific CD4 T cells[10,11]. Among people with HIV who developed seroprotective influenza antibody levels, lower CD154 expression of CD4 T cells[12] and lower PBMC IL-2 responses[13] were observed compared to controls. Although studies comparing CD8 T cell responses induced by licensed

[1]Institute of Clinical and Molecular Virology, Universitätsklinikum Erlangen, Friedrich-Alexander-Universität Erlangen-Nürnberg, Erlangen, Germany. [2]School of Infection & Immunity, University of Glasgow, Glasgow, UK. [3]Human Immunology Section, Vaccine Research Center, National Institutes of Health, Bethesda, USA. [4]Department of Medicine, Northwestern University, Feinberg School of Medicine, Chicago, USA. [5]Infectious Disease and Immunodeficiency Section, Department of Internal Medicine 3, Universitätsklinikum Erlangen, Friedrich-Alexander-Universität Erlangen-Nürnberg, Erlangen, Germany. [6]Department of Internal Medicine 3, Universitätsklinikum Erlangen, Friedrich-Alexander-Universität Erlangen-Nürnberg, Erlangen, Germany. ✉e-mail: krystelle.nganou@uk-erlangen.de

vaccines in people living with or without HIV are rare, the reduced capability of licensed vaccines to induce antibody and T cell-mediated immunity in HIV infection has been linked to impaired immune reconstitution. One of the most apparent measures of immune reconstitution in ART-treated people with HIV is the CD4 T cell recovery, that is improved under ART but fails to be normalized in most individuals[14]. ART-mediated recovery of the CD4 T cell count or the CD4:CD8 T cell ratio was shown to associate with a higher magnitude of post-vaccination antibody levels for influenza[15], hepatitis B[16,17], hepatitis A[18], tetanus and diphtheria toxoid[19] and with higher PBMC IFNγ responses after influenza vaccination[20]. While the initiation of ART was found to increase the proportion of recall CD4 T cell responses to tetanus toxoid[21] or to transiently increase lymphocyte proliferative responses to tetanus toxoid[22], CD4 T cell recovery alone could not explain the lower IFNγ responses or proliferative capacity of vaccinia virus- and *M. tuberculosis*-specific CD4 T cells in people with HIV[23]. Likewise, failure to induce protective antibody levels after influenza vaccination in up to 40% of people with HIV was reported to be independent of CD4 T cell recovery[24].

Aside from CD4 T cell count, other main features of immune recovery in HIV infection include the level of inflammation and immune activation that persist under ART, as measured by elevated blood concentration of several cytokine or chemokines along with higher cell surface expression of markers such as CD38 and HLA-DR on T cells[25–27]. In fact, inflammation and immune activation have been shown to favor disease progression, to limit immune reconstitution and to independently predict morbidities in people with HIV[28–31]. To date, there is a paucity of studies addressing the relationship between vaccine-induced immunity and persistent inflammation and immune activation in ART-treated people with HIV. Shive et al. reported that in the absence of ART, pre-vaccination plasma levels of IP-10 were associated with lower anti-tetanus toxoid antibody levels one week after vaccination[6]. Among ART-treated people with HIV, hepatitis B vaccine non-responders were reported to have higher CD8 T cell activation along with lower CD8 T cell IFNγ responses[5]. Moreover, post-vaccine levels of influenza antibodies were shown to inversely correlate with CD4 T cell activation that, in turn, positively correlated with pre-vaccine plasma levels of TNF[32]. While these studies together suggest an association between vaccine-induced immunity and inflammation and immune activation in HIV infection, data are limited, and underlying mechanisms remain unclear. This is particularly important to address in light of pandemic preparedness global initiatives that have been recently invigorated and involve vaccine-based strategies that would undoubtedly include people with HIV.

Here, we explored the maintenance of T cell-mediated immunity by comprehensively assessing recall responses to two independent antigens—measles virus (MV) and tetanus toxoid (TT)—in ART-treated people with HIV and uninfected persons with prior MV and TT immunity. We took a systems immunology approach to integrate T cell cytokine production and proliferation in response to MV and TT antigens, soluble and cellular markers of inflammation and immune activation, and transcriptome analysis of sorted antigen-stimulated T cells. Taken together, our findings reveal that loss of CD4 T cell responses to prior vaccines is associated with inflammation and immune activation, and with an overall impairment of antigen-induced T cell activation as well as downstream IFNγ signaling pathways in ART-treated people with HIV.

## Results
### Participants characteristics
For this study, 33 ART-treated people with HIV and 34 HIV-uninfected participants were recruited. All 67 participants were central Europeans with prior immunity established in childhood for tetanus toxoid and measles virus. A median duration of 4.5 years on ART resulted in complete suppression of plasma virus load along with increased CD4 T cell count and CD4:CD8 T cell ratio compared to the pre-ART time-point (Supplementary Fig. 1). At the start of this study, all participants

with HIV were on ART, had undetectable plasma virus load and a median CD4 T cell count of 748 cells/μl. For uninfected participants, self-reported absence of HIV infection was confirmed by a combined measurement of plasma HIV-1 antibodies, HIV-2 antibodies and p24 antigen. Detailed characteristics of study participants are presented in Supplementary Table 1.

### Lower TT and MV recall T cell responses in ART-treated people with HIV
To assess MV- and TT-specific recall T cell responses in our study participants, we first measured PBMC expression of CD69 as an activation marker after in vitro stimulation with the respective antigens. Here, the frequency of CD69$^+$ T cells following MV antigen stimulation was 2- to 6-fold lower in total CD3 as well as CD4 and CD8 T cell subsets of the HIV-infected group (Fig. 1A; $P \leq 0.002$). Following stimulation with TT antigen, the frequency of CD69$^+$ T cells was 2- to 4-fold lower in the HIV-infected group, albeit statistically significant for the CD8 T cell subset ($P = 0.03$) but not the CD4 T cell subset or the total CD3 T cell compartment ($P \geq 0.07$). Stimulation with the positive control staphylococcal enterotoxin B (SEB) generated potent CD69 expression in both groups, with markedly lower expression levels in the HIV-infected group ($P \leq 0.0001$). Analysis of cytokine production after MV and TT stimulation revealed a profile similar to the expression of CD69. For MV, the HIV-infected group had markedly lower cumulative expressions of IFNγ, IL-2 and TNF by total CD3 as well as CD4 and CD8 T cell subsets (Fig. 1B; $P \leq 0.0006$). A breakdown of these responses for individual cytokines is presented in Supplementary Fig. 2. For TT, the median expression of IFNγ, IL-2 and TNF was marginally lower but not statistically significant ($P \geq 0.42$). Stimulation with the SEB positive control induced strong and equally high cumulative cytokine expressions in both HU and HIV groups (Fig. 1B; $P \geq 0.07$), possibly related to the ability of superantigens to induce saturated cytokine production[33]. MV-, TT- and SEB-induced cytokine production within the fraction of CD69$^+$ T cells revealed a pattern similar to that of the total T cell fractions (Supplementary Fig. 3). Evaluation of CD8 T cell degranulation by measurement of CD107a expression did not reveal differences between the two groups (Fig. 1C; $P \geq 0.65$). Furthermore, measurement of Ki67 expression to assess T cell proliferative capacity showed that compared to uninfected persons, the HIV-infected group had a lower frequency of Ki67$^+$ T cells in response to stimulation with MV (Fig. 1D; $P = 0.0002$) but not TT ($P = 0.41$). In multivariate analysis, MV-specific IFNγ and CD69 responses of all T cell subsets as well as frequency of Ki67$^+$ T cells remained lower in the HIV-infected group after correction for multiple testing (Supplementary Table 2). Altogether, these data indicated a loss of antigen-specific T cell responses, particularly MV-specific CD4 T cell responses, in ART-treated people with HIV.

### Persistent inflammation, immune activation and altered composition of T cell subsets associate with lower recall T cell responses
Next, soluble markers of inflammation and immune activation were measured in our study participants. Compared to uninfected persons, elevated plasma concentrations were observed in the HIV-infected group for CXCL13 ($P = 0.01$), IFNγ ($P < 0.0001$), IL-4 ($P < 0.0001$), IL-6 ($P = 0.0005$), IL8 ($P < 0.0001$), IL-10 ($P = 0.0003$.), IL-12 ($P = 0.002$) and TNF ($P < 0.0001$) but not for C-reactive protein (CRP) or MIP1β (Fig. 2A). These higher levels of cytokines and chemokines typically produced by various immune cells in response to endogenous or exogenous triggers confirmed an overall state of inflammation in the HIV-infected group despite effective ART and complete suppression of viral loads in the blood. Moreover, higher plasma concentrations of sCD14 ($P = 0.0005$), sCD163 ($P = 0.04$) and IFABP ($P = 0.0002$) but not LBP were also observed in the HIV-infected group (Fig. 2B). While higher levels of sCD14 and sCD163 indicated monocyte and macrophage activation, elevated plasma concentrations of IFABP suggested

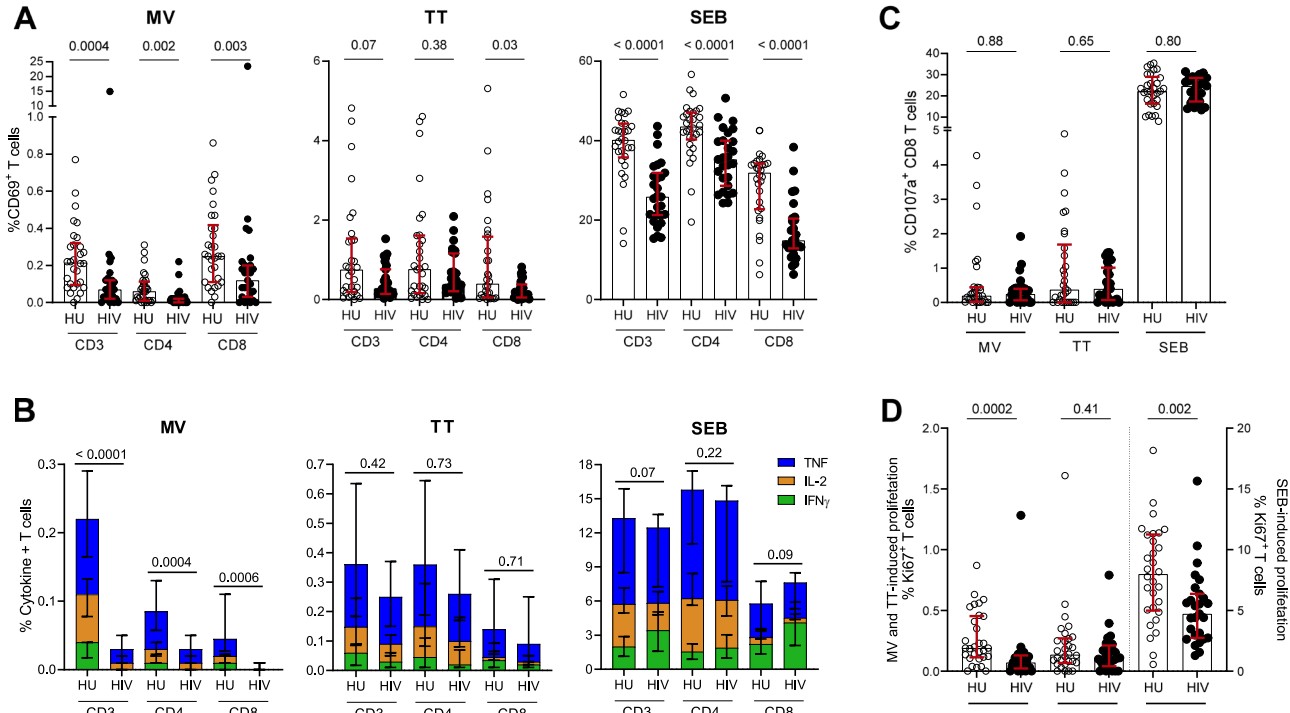

**Fig. 1 | Recall T cell responses to prior vaccine antigens are lower in ART-treated people with HIV. A** Percentages of CD69+ CD3, CD4 or CD8 T cells after stimulation of PBMCs from uninfected (HU) or HIV-infected (HIV) groups with MV, TT or SEB. **B** Percentages of IFNγ+, IL-2+ and TNF+ CD3, CD4 or CD8 T cells after stimulation with MV, TT or SEB. **C** Percentages of CD107+ CD8 T cells after stimulation MV, TT or SEB. **D** Percentages of Ki67+ CD3 T cells after stimulation with MV, TT or SEB. For each outcome measure, responses of T cell subsets to MV, TT or SEB are presented after subtraction of the respective unstimulated background for each participant. Circles represent individual participants (30 HU; 27 HIV for panels **A**–**D**), medians and interquartile range (IQR—See also Supplementary Figs. 2 and 3) are shown, comparison between groups was performed by two-tailed Mann–Whitney tests and *P*-values of comparisons are presented. Source data are provided as a Source Data file.

ongoing microbial translocation, that is known to contribute to chronic inflammation by skewing differentiation of immune cells[34]. Further analysis of the T cell expression of HLA-DR, PD-1 and CD57, that are increased in people with HIV and were linked to impaired T cell dysfunction[35–37], showed higher expression of HLA-DR, PD-1 and CD57 predominately on CD4 T cells in the HIV-infected group (*P* ≤ 0.015; Fig. 2C), denoting CD4 T cell activation and exhaustion. In the CD8 T cell subset, only HLA-DR expression was different between the groups with higher levels in the HIV-infected group (*P* = 0.04; Fig. 2C). No difference was observed for percentages of CD56hi NK cells or monocytes (Fig. 2C). In multivariate analysis, the frequencies of CD57+, HLA-DR+ or PD1+ CD4 T cells as well as plasma levels of various markers, namely CXL13, IFNγ, IFABP, IL-4, IL-6, IL-8, IL-10, IL-12, sCD14 and TNF remained higher in the HIV-infected group after correction for multiple testing (Supplementary Table 3). With the exception of HLA-DR+ CD4 T cells, these measures were also higher in CMV seropositive people with HIV compared to CMV seropositive HIV-uninfected persons (Fig. 2D). Thus, despite undetectable plasma HIV load, improved CD4 T cell count and CD4:CD8 T cell ratio, persistent inflammation, CD4 T cell hyperactivation and senescence was observed in the HIV group.

We next explored the relationship between the observed lower recall T cell responses to prior vaccine antigens and the persistence of inflammation and immune activation. A dichotomous segregation characterized by a high inflammation profile alongside lower T cell responses was noticeable in the HIV-infected group, while the uninfected group had the inverse profile (Fig. 3A; Supplementary Fig. 4). Spearman correlation analysis further showed that over our entire study cohort, the various markers of inflammation and immune activation positively correlated with each other. Likewise, all measures of T cell responses to prior vaccine antigens also positively correlated with each other. In contrast, relationships between inflammation and recall T cell responses to antigens were dominated by inverse correlations (Fig. 3B–D). For example, MV-specific T cell responses inversely correlated with circulating levels of CXCL13, IFABP, IFNγ, IL-4, IL-6, IL-8, IL-10, TNF, sCD14, sCD163, CD4 HLA-DR+, CD4 PD-1+, CD4 CD57+ and CD8 CD57+ while these only positively correlated to one another. Higher PBMC mRNA levels of *SOCS1* and *SOCS3* but not *FOXP3* or *GATA3* in the HIV-infected group suggested that plasma cytokines may have a direct influence on T cell responses through suppressor of cytokine signaling proteins (Fig. 3E).

Detailed measurement of naïve and memory T cell subsets showed that the HIV-infected group had lower frequencies of central memory (CM) but higher frequencies of stem cell memory (SCM) and effector memory (EM) CD4 T cells, particularly due to an enrichment of EM intermediate and less EM early or terminal cells (Fig. 4A). Frequencies of Th1, Th2 and naïve CD4 T cells were not different between HIV-infected and uninfected groups. CD8 T cells of the HIV-infected group showed less naïve cells but more effector and EM cells, driven by an enrichment of terminal over early and intermediate cells (Fig. 4B). Moreover, the expression of CD57 was higher on Th1, Th2, CD4 EM, CD4 CM and CD8 CM T cells of the HIV-infected group (Supplementary Fig. 5A). Similar to indicators of inflammation and immune activation, the frequencies of subsets that where enriched in the HIV-infected group negatively correlated with recall T cell responses whereas the frequencies of depleted subsets positively correlated with recall T cell responses (Fig. 4C, Supplementary Fig. 5B, C). Taken together, these data highlighted that in ART-treated people with HIV, persistent inflammation and immune activation as well as altered composition of the T cell compartment as seen by an enrichment of more differentiated cells, associated with lower T cell responses to prior vaccine antigens.

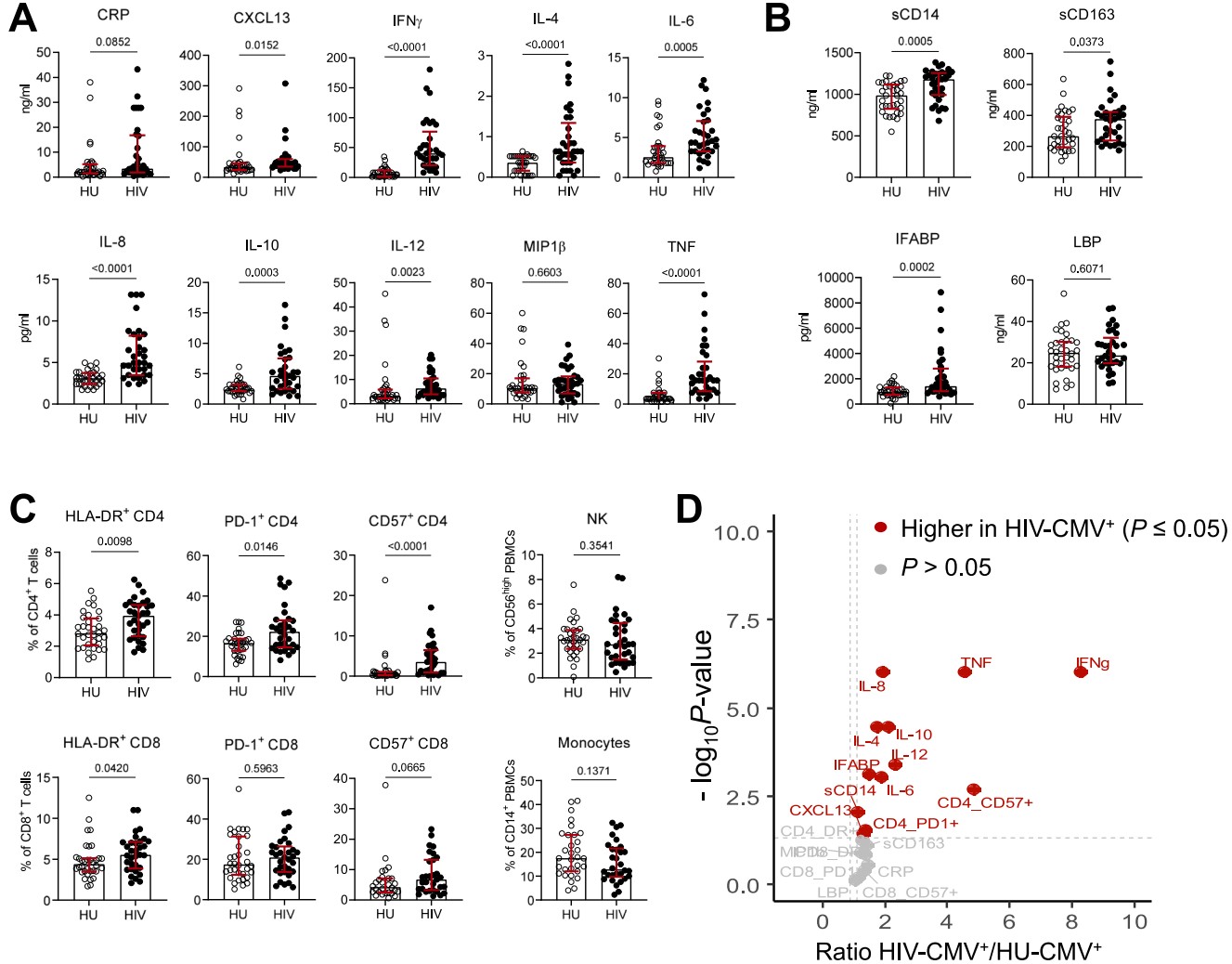

**Fig. 2 | Higher levels of soluble and cellular markers of inflammation and immune activation in ART-treated people with HIV. A** Plasma levels of CRP, CXCL13, IFNγ, IL-4, IL-6, IL-8, IL10, IL12, MIP1β and TNF. **B** Plasma levels of sCD14, sCD163, IFABP and LBP. **C** Blood frequencies of HLA-DR⁺, PD-1⁺ and CD57⁺ CD4 or CD8 T cells as well as frequencies of CD56high NK cells and monocytes. Circles represent individual participants (33 HU; 34 HIV), medians and IQR are shown,

comparison between groups was performed by two-tailed Mann–Whitney tests and *P*-values of comparisons are presented. **D** Fold differences of univariate comparisons for plasma markers and cell frequencies between HIV-CMV seropositive and HU-CMV seropositive participants as determined by two-tailed Mann–Whitney tests without adjustment for multiple comparison. Source data are provided as a Source Data file.

## HIV-infected and uninfected groups showed distinct global transcriptome profile of antigen-specific CD4 T cells

Additional stimulation assays with MV and TT antigens were then performed with 17 HIV-infected and 18 uninfected participants prior to sorting various cell subsets for analysis of gene expression by RNA sequencing. Median levels of all prior measures were similar between the total HIV-infected or uninfected group and the respective randomly chosen participants for transcriptome analysis (Supplementary Table 4). Because our prior observations of higher T cell activation and lower antigen-specific T cell responses primarily affected the CD4 T cell compartment, we next investigated transcriptional changes specifically in the following CD4 T cell subsets: (1) MV or TT antigen-specific CD4 T cells defined as CD45RO⁺ CD69⁺ CD154⁺ and henceforth simply referred to as CD154⁺; (2) MV or TT stimulated total memory CD4 T cells that are not antigen-specific defined as CD45RO⁺ CD69⁻ CD154⁻ and further referred to as CD154⁻; (3) unstimulated memory CD4 T cells that are defined as CD45RO⁺ and also lacked expression of either CD69 or CD154. Of note, CD154 expression has been shown to be a reliable marker to identify antigen-specific CD4 T cells[38]. Our gating strategy is shown in Supplementary Fig. 6. To exclude the possibility that a potential reactivation of HIV in infected cells may have contributed to the loss of antigen-specific

T cells during stimulation, although unlikely due to the relatively short stimulation, copies of HIV gag were measured in sorted CD4 T cells. With up to 47.1% of the analyzed participants having detectable HIV gag copies, there were no significant differences in the HIV gag copies or the proportion of HIV gag positive CD154⁺ or CD154⁻ CD4 memory T cells before or after MV or TT stimulation (Supplementary Fig. 7).

Following RNA sequencing of sorted subsets, Principal Component Analysis of normalized counts from the gene expression analysis showed a clustering mainly driven by the cell subset, with CD154⁺ CD4 T cells segregating from CD154⁻ CD4 T cells irrespective of the stimulation condition (Supplementary Fig. 8A), along with a high expression of CD4, a low expression of CD8, and higher expression of CD154 and CD69 was observed in the CD154⁺ subset only (Supplementary Fig. 8B). Based on the heatmaps of significantly differential genes (DEGs) across subsets, distinct genes were up- or downregulated in stimulated conditions compared to the unstimulated condition; and this observation was consistent within HIV-infected or uninfected groups for both MV and TT stimulations (Supplementary Fig. 8C). Differential gene expression analysis allowed grouping of genes with similar expression profiles across samples where in for both MV and TT stimulations, we identified 6 signature profiles of gene expression,

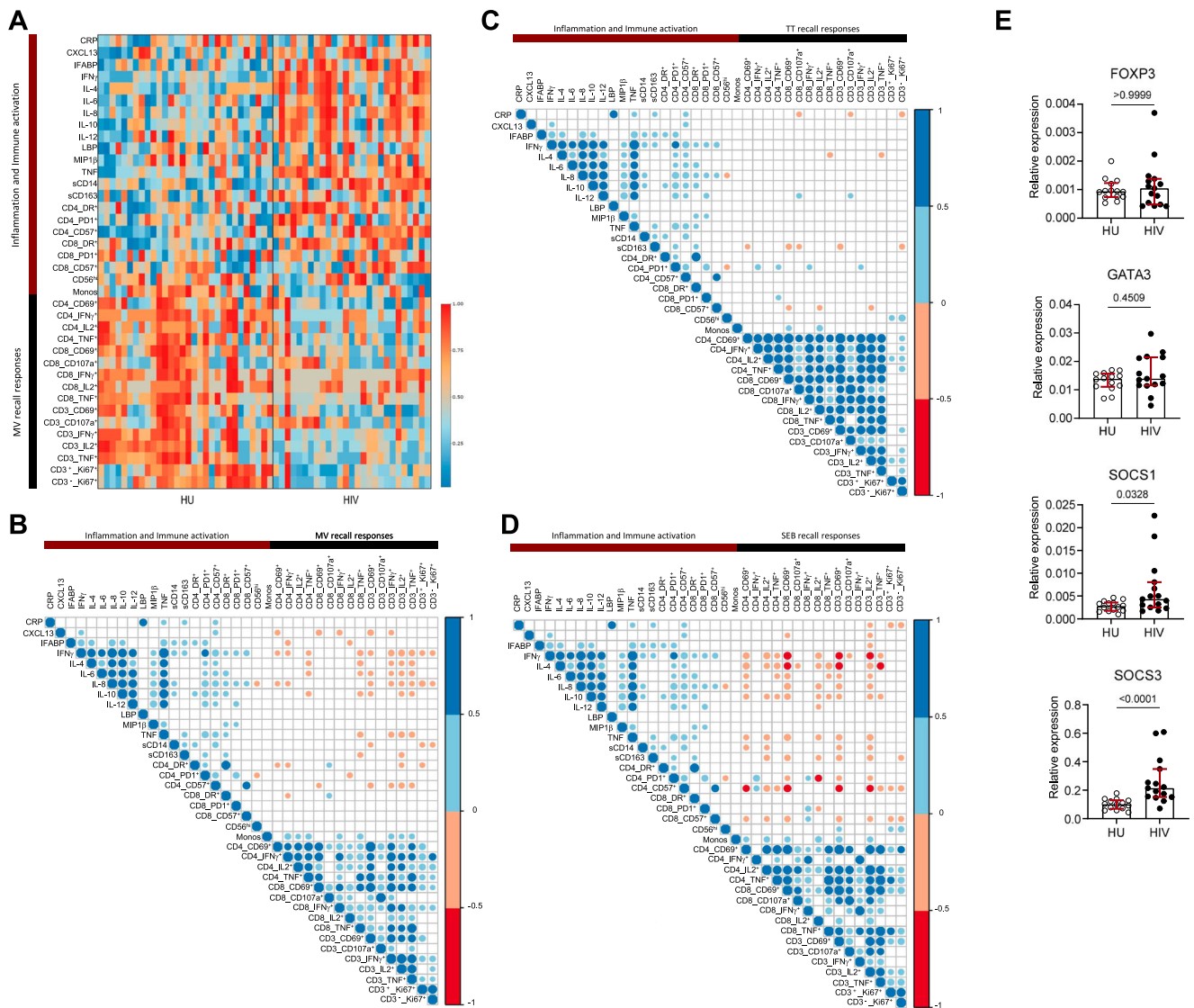

**Fig. 3 | Higher levels of soluble and cellular markers of inflammation and immune activation in ART-treated people with HIV correlate with lower recall T cell responses. A** Heatmap integrating all inflammation and immune activation measures with in vitro T cell responses to MV stimulation. Percentized expression levels across donors are presented for each variable and colors indicate the highest (red) or lowest (blue) expression level for a given marker. **B–D** Correlogram of T cell responses after MV (**D**), TT (**C**) or SEB (**D**) stimulations in relation to inflammation and immune activation measures. Colored circles represent correlations with

$P \le 0.05$ as determined by Spearman analysis. Blue and red circles indicate positive and negative correlations, respectively. Color intensity and the size of the circle are proportional to the correlation coefficients. The corresponding colors of the correlation coefficients are presented in the legend. **E** PBMC mRNA levels of *FOXP3*, *GATA3*, *SOCS1* and *SOCS3*. Circles represent individual participants (16 HU; 17 HIV), medians and IQR are shown, *P*-values of two-tailed Mann–Whitney comparisons between groups are presented. Source data are provided as a Source Data file.

each containing unique genes with distinct distribution across subsets and stimulation conditions in both HIV-infected and uninfected groups (Supplementary Fig. 8D, E). Further comparison of the number and directionality of DEGs between subsets for a given group showed variability in the number of up or downregulated genes. Following MV or TT stimulation, 108 to 471 DEGs were found for the CD154⁻ subset, while a higher number of DEGs ranging between 1868 and 2277 were observed for CD154⁺ cells (Fig. 5A). In a comparative analysis of the overlap of these DEGs between the HIV-infected and uninfected groups, several DEGs appeared in the HIV-infected but not the uninfected group, and vice versa after both MV and TT stimulations (Fig. 5B; Supplementary Data 1). Pathways analysis of these unique DEGs showed that pathways with *P*-adj ≤ 0.05 in the HIV-uninfected group included production and regulation of type I IFN, regulation of leukocyte activation and mononuclear cell differentiation that were not found in the HIV-infected group. (Fig. 5C). Thus, while the global

transcription profile between HIV-infected and uninfected was similar after TT and MV stimulation, with up- or downregulation of distinct genes and the largest number of DEGs in antigen-specific CD4 T cells, we detected numerous DEGs that were unique to either HIV-infected and uninfected groups.

**Vaccine-specific gene signatures and IFN signaling pathways were downregulated in the HIV-infected group**

To unravel unique transcriptional features that discriminated between HIV-infected and uninfected groups and could explain the lower recall CD4 T cell responses to prior vaccines observed in the HIV-infected group, we next performed differential expression analysis between groups at the individual gene level. Of almost 20,000 genes, we identified across the various subsets and stimulation conditions 555 to 733 genes that were significantly up- or downregulated in the HIV-infected compared to the uninfected group at cutoff *P* < 0.05, but

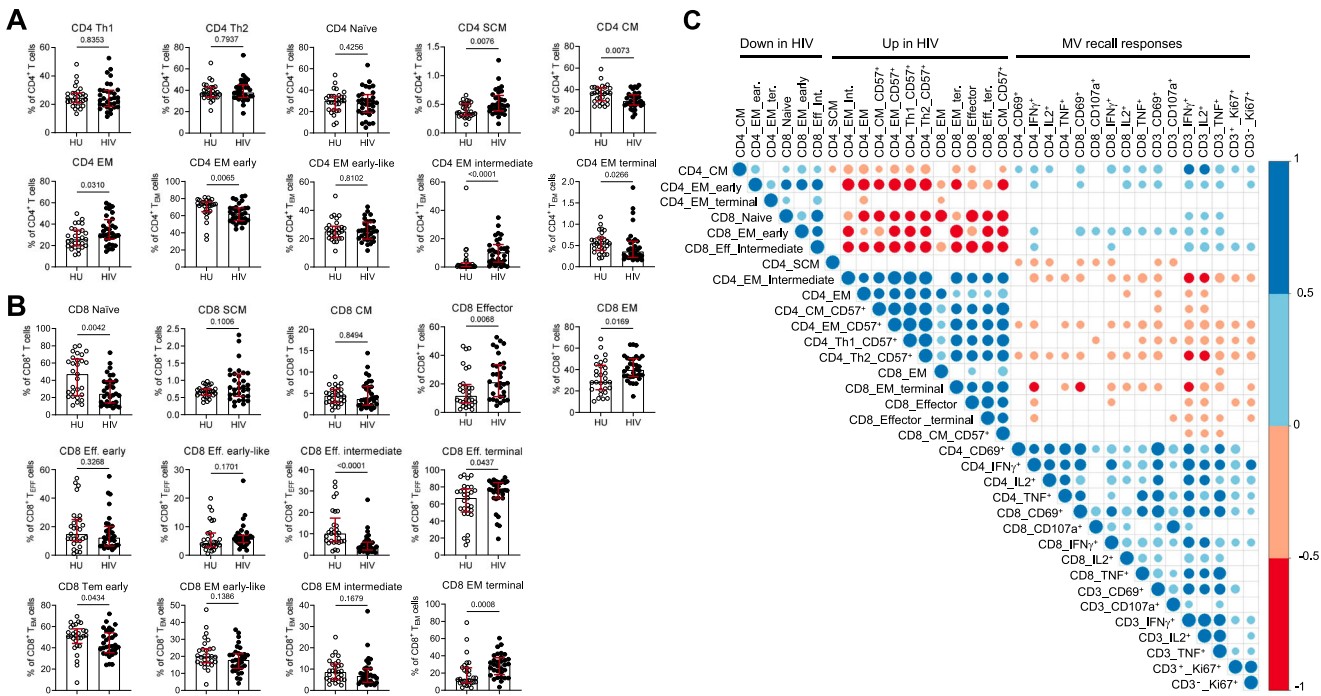

**Fig. 4 | Altered composition of the naïve and memory T cell subset compartment in ART-treated people with HIV. A** Frequencies of CD4 T cell subsets. **B** Frequencies of CD8 T cell subsets. Circles represent individual participants (29 HU; 32 HIV), medians and IQR are shown, *P*-values of two-tailed Mann–Whitney comparisons between groups are presented. CM central memory, EM effector memory, SCM stem cell memory. **C** Correlogram of T cell responses after MV

stimulation in relation to T cell subsets frequencies. Colored circles represent correlations with *P* ≤ 0.05 as determined by Spearman analysis. Blue and red circles indicate positive and negative correlations, respectively. Color intensity and the size of the circle are proportional to the correlation coefficients. The corresponding colors of the correlation coefficients are presented in the legend. Source data are provided as a Source Data file.

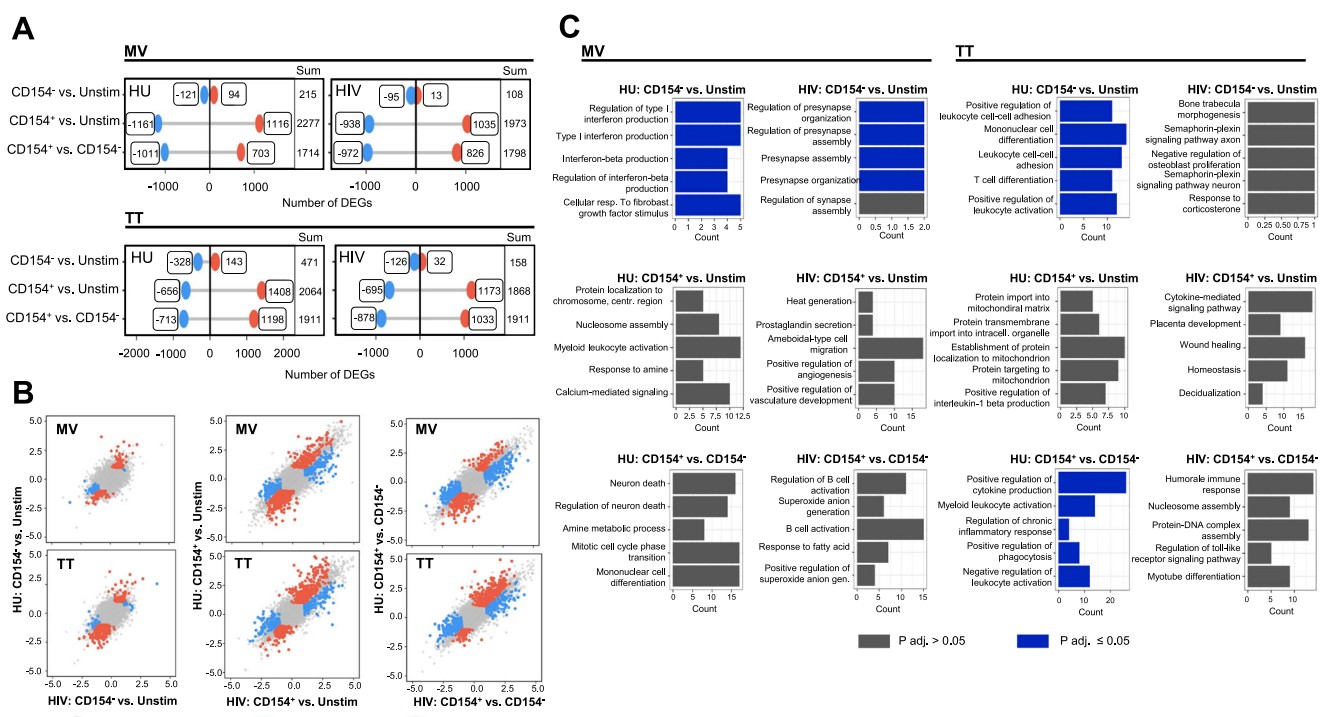

**Fig. 5 | The transcriptional landscape of antigen-specific and memory CD4 T cells changes upon MV and TT stimulation in both HIV-infected and uninfected groups. A** Lollipop plots of the number and directionality of differentially expressed genes (DEGs) between subsets after MV and TT stimulation. Red and blue color indicate up and downregulation, respectively. **B** Overlap scatterplots of DEGs between subset conditions for uninfected and HIV-infected groups. Red and

blue dots represent DEGs that are significantly enriched only in the HIV-infected or uninfected group, respectively. Gray dots represent genes that are either not significant within either group or shared between groups. **C** Pathway analysis of DEGs unique to the HIV-infected or uninfected groups. Pathways with *P*-adj ≤ 0.05 are presented in blue. Adjustment for multiple comparisons was performed by Benjamini–Hochberg correction.

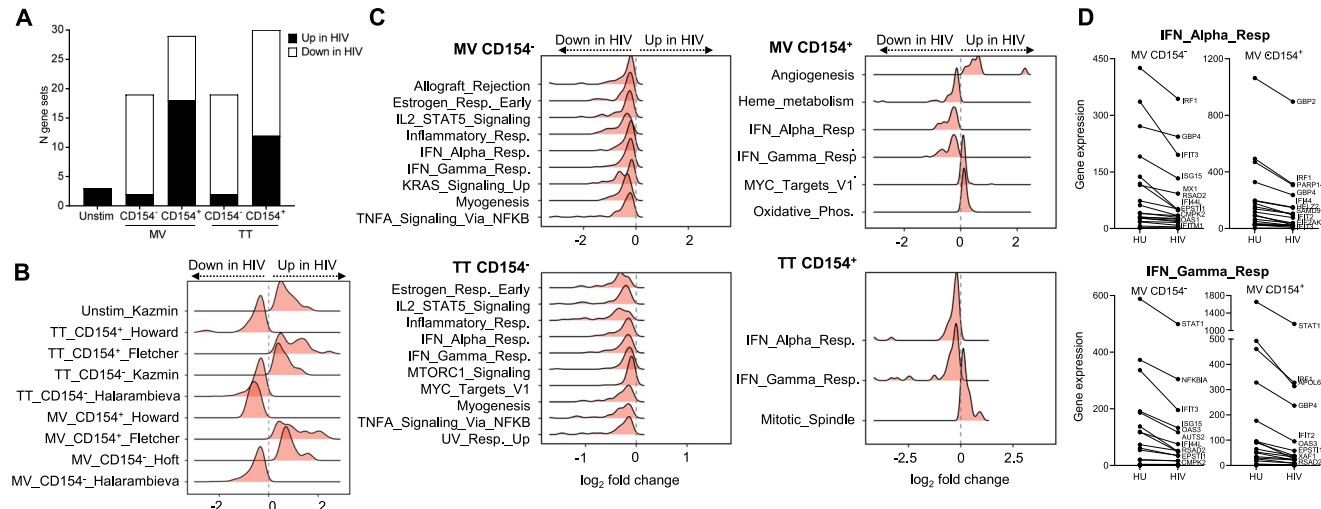

**Fig. 6 | Gene set enrichment analysis of sorted MV or TT-stimulated CD4 T cell subsets.** **A** Number of MSigDB gene sets up or downregulated in HIV-infected compared to uninfected for each T cell subset (CD154⁻ or CD154⁺) and antigen (MV or TT). **B** Ridgeline plot showing the distribution of genes fold changes between HIV-infected and uninfected groups for the most enriched VAX gene set in either group. **C** Ridgeline plots of gene fold changes between HIV-infected and uninfected groups within the MSigDB HALLMARK (H) pathways significantly enriched in either group for each T cell subset and antigen. **D** Dot plots of the top 20 most down-regulated genes by Log_2_Fold of the HALLMARK_IFN_ALPHA_RESPONSE and HALLMARK_IFN_GAMMA_RESPONSE pathways in MV-stimulated CD4 T cells (18 HU; 17 HIV). Source data are provided as a Source Data file.

almost none passed the *P*-adj cutoff (Supplementary Data 2). It is known that differential expression analysis requires a strict cutoff in order to classify genes as up- and downregulated, and computes *P*-values assuming the genes are independent of each other, which does not necessarily align with biological mechanisms involving intimately linked expression of several genes. On the other hand, gene set enrichment analysis (GSEA) focuses on the pathway rather than the gene and allows identification of pathways that may explain biological relevant functions associated with a list of genes[39]. Therefore, we performed GSEA of our dataset using the MSigDB VAX dataset, a subcollection of the C7 immunologic signature gene sets that contains curated gene expression results of responses to various vaccines, where experimental approaches included in vitro stimulation with vaccine antigens. Within the VAX dataset, up to 30 gene sets were significantly enriched (*P*-adj < 0.05) in either HIV-infected or uninfected groups with the CD154⁺ subsets yielding the largest number of differential gene sets (*N* = 30) and the unstimulated condition yielding the lowest number of differential gene sets (*N* = 3; Fig. 6A). A table including enrichment scores and adjusted *P*-values of all VAX gene sets significantly enriched in either HIV-infected or uninfected groups is provided in Supplementary Data 3. For each of the five subset conditions, the gene sets with highest *P*-adj and normalized enrichment score above or under zero, thus showing the strongest enrichment in either HIV-infected or uninfected, were chosen for a more detailed description (Table 1). The ´Howard´ gene set comprising upregulation of cytokine signaling, in particular IFN signaling, MHC mediated antigen presentation and IFN-gamma signaling was previously shown to be upregulated after influenza vaccination[40] but was downregulated in both MV-specific and TT-specific CD154⁺ T cells of our HIV cohort (Fig. 6B). Concomitantly, these subsets showed higher expression of the ´Fletcher´ gene set (Fig. 6B) identified after purified protein derivative (PPD) stimulation of PBMCs from *M. tuberculosis* vaccinated persons, where pro-inflammatory macrophage response and downregulation of anti-inflammatory responses were main features[41]. In the MV- and TT-stimulated CD154⁻ T cells of the HIV-infected group, we observed a downregulation of the ´Halarambieva´ gene set (Fig. 6B) that includes genes for cell adhesion and migration, cytokine/chemokine activity and were upregulated upon MMR vaccination[42]. In the TT-stimulated CD154⁻ T cells of the HIV-infected group, there was an

upregulation of the ´Kazmin´ gene set (Fig. 6B) previously described to be downregulated by the malaria vaccine RTS,S; for which the induced protection was linked to genes for TLR signaling, IFN I responses or antigen presentation[43]. In the MV-stimulated CD154⁻ T cells of the HIV-infected group, there was an upregulation of the ´Hoft´ gene set (Fig. 6B) that includes, among others, NK cell activation pathways previously found to negatively correlate with immunogenicity of a recombinant Bacillus Calmette–Guérin (BCG) vaccine[44]. Thus, across the 5 subsets conditions, the HIV-infected group showed a reduced expression of gene sets that were previously reported to associate with protective immune responses as well as an enrichment for gene sets that were previously reported to negatively associate with vaccine-induced protective immunity.

In order to further clarify the various cellular pathways that were altered in the HIV-infected group, we performed GSEA using the MSigDB Hallmark dataset that summarizes well-defined biological processes[45]. For the most part, we observed a prominent downregulation of Hallmark pathways in all TT and MV-stimulated CD4 T cell subsets of the HIV-infected group, whereas no Hallmark pathway could discriminate between the groups for the unstimulated condition. Irrespective of the antigen, downregulated pathways in the CD154⁻ subset included TNF signaling via NFκB, the IL-2/STAT5 signaling pathway and the inflammatory response signaling pathway (Fig. 6C). In MV and TT-stimulated CD154⁻ and CD154⁺ subsets, the IFNα and IFNγ response pathways were consistently downregulated (Fig. 6C, D; Supplementary Fig. 9), aligning with the in vitro cytokine responses and findings from the VAX dataset analysis. Normalized enrichment scores and adjusted *P*-values of the Hallmark pathways with differential expression between groups are presented in Supplementary Data 4.

**Low recall responses are linked to distinct transcriptomic signatures and can be rescued by anti-inflammatory drugs in vitro**
Lastly, we performed an integrative analysis of our data combining plasma and cellular markers of inflammation and immune activation, MV or TT recall responses and gene expression of MSigDB Hallmark pathways of MV or TT-stimulated memory CD4 T cells. Here, we included all pathways that were downregulated in CD154⁻ or CD154⁺ memory CD4 T cells of the HIV-infected group (as shown in Fig. 6). The integrative analysis revealed that irrespective of the antigen, levels of

**Table 1 | Top MSigDB VAX gene sets up- or downregulated in HIV-infected compared to uninfected**

| | MSigDB VAX gene sets | P-adj | NES | Size | LE |
|---|---|---|---|---|---|
| | **Unstimulated CD154neg HIV-infected vs uninfected** | | | | |
| Up in HIV | KAZMIN_PBMC_P_FAL._RTSS_AS01_AGE_UNKNOWN_CORR._WITH_PROTECTION_56DY_NEG. | 1.9E-05 | 0.68 | 2.13 | 41 |
| Down in HIV | None | | | | |
| | **MV CD154neg HIV-infected vs uninfected** | | | | |
| Up in HIV | HOFT_PBMC_TICE_BCG_RBCG_AG85A_AG85B_AGE_18_40YO_CORR._W_W_BL._BAC._ACT._NEG. | 9.0E-05 | 0.62 | 1.97 | 40 |
| Down in HIV | HARALAMBIEVA_PBMC_M_M_R_II_AGE_11_22YO_VACCINATED_VS_UNVACCINATED_7YR_UP | 5.3E-11 | −0.45 | −1.63 | 752 |
| | **MV CD154pos HIV-infected vs uninfected** | | | | |
| Up in HIV | FLETCHER_PBMC_BCG_10W_INFANT_PPD_STIMULATED_VS_UNSTIMULATED_10W_UP | 1.6E-08 | 0.67 | 2.14 | 83 |
| Down in HIV | HOWARD_T_CELL_INACT_MONOV_INFL._A_INDONESIA_05_2005_H5N1_AGE_18_49YO_1DY_UP | 1.1E-07 | −0.75 | −2.31 | 42 |
| | **TT CD154neg HIV-infected vs uninfected** | | | | |
| Up in HIV | KAZMIN_PBMC_P_FAL._RTSS_AS01_AGE_UNKNOWN_CORR._WITH_PROTECTION_56DY_NEG. | 3.5E-4 | 0.63 | 1.89 | 41 |
| Down in HIV | HARALAMBIEVA_PBMC_M_M_R_II_AGE_11_22YO_VACCINATED_VS_UNVACCINATED_7YR_UP | 5.3E-13 | −0.42 | −1.77 | 752 |
| | **TT CD154pos HIV-infected vs uninfected** | | | | |
| Up in HIV | FLETCHER_PBMC_BCG_10W_INFANT_PPD_STIMULATED_VS_UNSTIMULATED_10W_UP | 5.7E-10 | 0.63 | 2.47 | 83 |
| Down in HIV | HOWARD_DENDR._CELL_INACT_MONOV_INFL._A_IND._05_2005_H5N1_AGE_18_49YO_1DY_UP | 7.1E-07 | −0.60 | −1.73 | 141 |

NES normalized enrichment score, LE leading edge.

Adjustment for multiple comparisons was performed by Benjamini–Hochberg correction.

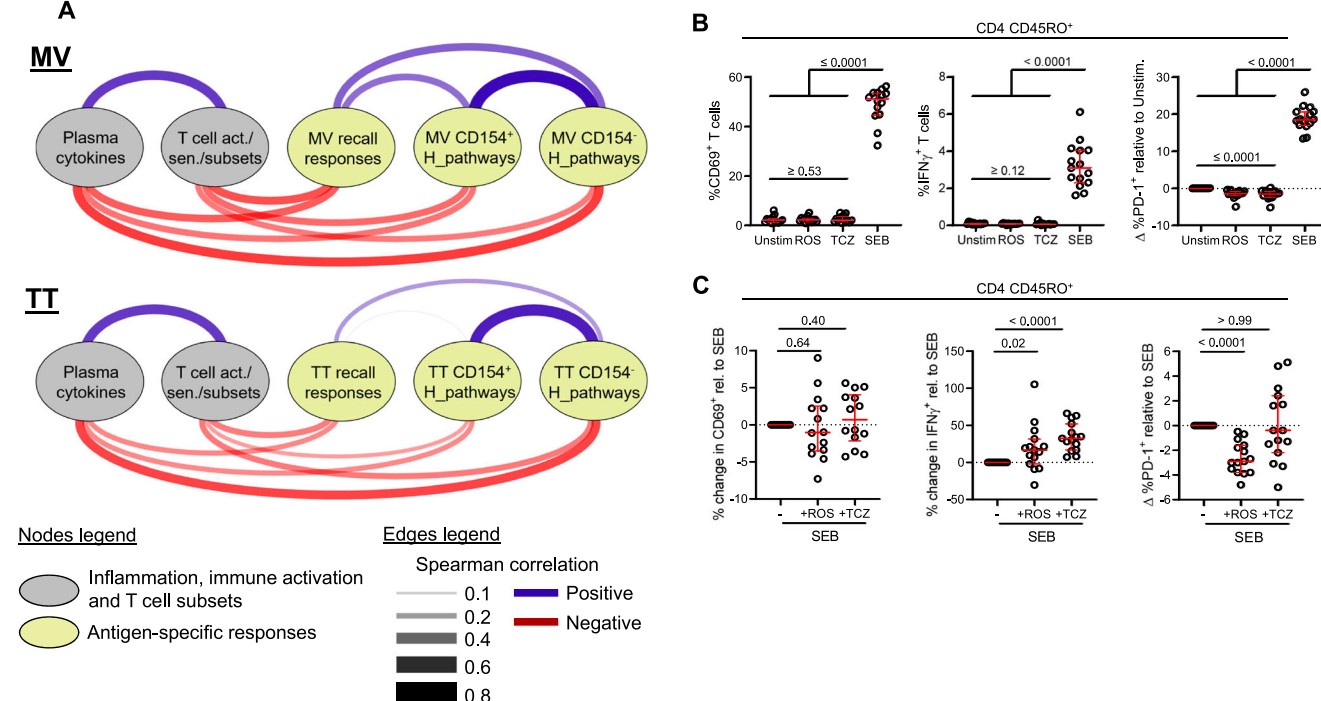

**Fig. 7 | Integrative analysis of plasma, cell markers, cell subsets and transcriptome of MV or TT-stimulated CD4 T cell subsets. A** Arc diagram of correlations between plasma cytokines, T cell markers and subsets enriched in the HIV-infected group, recall responses and Hallmark pathways across HIV-infected and uninfected groups. Color of the edges between nodes indicates Spearman correlations. Color intensity and the size of the edges are proportional to the correlation coefficients. **B** Percentages of CD69+, IFNγ+ and difference in percentages of PD1+ memory CD4 T cells of 15 ART-treated HIV people upon stimulation with rosuvastatin (ROS), tocilizumab (TCZ) or SEB compared to the unstimulated condition. **C** Difference in percentages of CD69+, IFNγ+ and PD1+ memory CD4 T cells of 15 ART-treated HIV people upon stimulation with SEB+ROS or SEB+TCZ compared to SEB alone. Medians and interquartile range are shown, and comparisons between unstimulated, SEB, ROS or TCZ conditions were performed by two-tailed Wilcoxon paired test. Source data are provided as a Source Data file.

markers of inflammation and immune activation were inversely correlated with recall responses and with the Hallmark pathways (Fig. 7A). This observation suggested that inflammation and immune activation may alter cellular response pathways, including the IFNα and IFNγ response pathways of antigen-specific CD4 T cells, thereby resulting in loss of CD4 T cell immunity to prior vaccines. To validate our findings of an association between inflammation and CD4 T cell immunity and, at the same time, explore the potential effects of targeting inflammation, we next stimulated PBMC of 15 ART-treated HIV people with SEB alone or in combination with rosuvastatin (ROS) or tocilizumab (TCZ). In clinical trials with people with HIV, both drugs lowered indices of inflammation such as plasma sCD14 but not T cell activation for ROS, whereas TCZ lowered plasma sCD14, sCD40L, D-dimer, IL-7, sTNFR-1 and P-selectin[46,47]. Here, SEB was chosen as antigen because SEB-induced T cell responses were linked to inflammation (Fig. 3), and the toxin could induce measurable intracellular IFNγ in all participants,

thus providing a more sensitive readout compared to MV or TT antigens. PBMC stimulation with SEB alone induced high expression of CD69, IFNγ and PD-1 on memory CD4 T cells. Treatment with ROS or TCZ alone had no effect on CD69 or IFNγ expression, but PD-1 levels were lowered by both drugs (Fig. 7B). Moreover, pre-treatment with TCZ and ROS starkly increased the percentage of IFNγ+ memory CD4 T cells in SEB-stimulated cells despite undetectable effects on CD69 expression (Fig. 7C). Pre-treatment with ROS also lowered SEB-induced expression of PD-1 on memory CD4 T cells. Of note, ROS- and TCZ-mediated increase of IFNγ responses to SEB was also observed in CD8 and CD3 memory T cells along with a lowered PD-1 expression by ROS only (Supplementary Fig. 10). We also assessed whether recall T cell responses of ART-treated people with HIV could be improved by inhibitors of glycogen synthase kinase 3 (GSK3), that could interfere with metabolic functions supporting T cell differentiation. Our assays, however, showed no beneficial effect of the GSK3 inhibitors CHIR-99021 and SB415286 on T cell recall responses (Supplementary Fig. 11). Overall, these findings further supported the notion that in people with HIV, persistent inflammation limits recall T cell responses that can be rescued by anti-inflammatory drugs.

## Discussion

This study aimed to clarify whether persistent inflammation and immune activation in ART-treated HIV infection influences T cell responses to prior vaccines and to identify underlying mechanisms. Our work demonstrated that upon exposure to vaccine antigens, specifically TT and MV, ART-treated people with HIV had lower CD69 expression and lower frequencies of IFNγ/IL2/TNF+ T cells, where MV-specific CD4 T cell responses were the most affected. These findings were complemented by observations of an altered transcriptional profile of CD4 T cells in the HIV-infected group, that showed a downregulation of the IFNα and IFNγ signaling pathways in both non-responsive (i.e., CD154−) and responsive (i.e., CD154+) memory CD4 T cell subsets along with additional downregulation of among others IL2-STAT5, Myc, mTORC1 and Kras signaling pathways in non-responsive memory CD4 T cells. A large body of literature has shown that the IL2-STAT5 signaling pathway plays a major role in the regulation of CD4 T cell differentiation, effector function, metabolism and survival[48]. Upon IL-2-mediated T cell activation, IL-2 induces Blimp-1 through STAT5, thereby driving Th1 cell differentiation while simultaneously activating the metabolic regulator mTOR[49,50]. Activation of mTOR signaling in effector T cells further promotes protein synthesis, cell growth, clonal expansion and guides Th1 cell expression of the transcription factor T-bet that is necessary for the production of IFNγ[51–53]. Moreover, STAT5 has been reported to be a master regulator of energy and amino acid metabolism in CD4 T helper cells in order to accommodate for the high energy demand of T cell activation and proliferation, characterized by IL-2–mediated mTOR signaling and promoted Myc-driven metabolic changes[54]. Thus, our observation of lower IL2-STAT5, Myc, and mTORC1 signaling pathways may indicate a reduced ability to initiate T cell activation upon antigen encounter in ART people with HIV. Of note, KRAS is a key regulator of cell proliferation, differentiation and survival, and *Kras* deficiency has been shown to reduce T cell proliferation[55]. Because we observed lower expression of IL2-STAT5, Myc, mTORC1 and Kras signaling pathways in stimulated CD154− CD4 T cells but not in stimulated CD154+ CD4 T cells, we postulate that lower antigen-specific CD4 T cell responses were likely attributed to failure to induce T cell activation in a subset of cells that remained non-responsive whereas a second subset of cells may have succeeded at least in part to induce T cell activation but remained functionally impaired. In the latter, lower gene expression of the IFNα and IFNγ signaling pathways likely translated to lower CD69 surface expression and/or cytokine production.

Previous studies reporting lower frequencies of *M. tuberculosis*-specific CD4 T cells and lower cytokine responses to *M. tuberculosis* and *C. albicans* but not cytomegalovirus (CMV) in people with HIV[56,57],

along with preferential HIV in vitro infection of *C. albicans*- but not CMV-specific CD4 T cells of HIV-uninfected participants[58] have suggested that HIV contributes to a vulnerable adaptive immunity to some pathogens due to a loss of antigen-specific CD4 T cells. Our work demonstrated that irrespective of frequencies, antigen-specific CD4 T cells of ART-treated people with HIV exhibited transcriptional and functional differences that were linked to persistent inflammation and impacted on immunity to prior vaccines. Plasma levels of IL-4, IL-6, IL8 and IFNγ, among others, were higher in the HIV-infected group in a multivariate analysis and concentrations of these cytokines inversely correlated with antigen-induced T cell activation, cytokine production or proliferation upon MV and SEB stimulations. In contrast to MV and TT, the larger pool of antigen-specific T cells induced by SEB likely allowed for more sensitive detection of the relationships between recall T cell responses and inflammation and immune activation. SEB triggers polyclonal T cell activation upon obligatory binding to the TCR and has been shown to induce strong activation and responses of antigen-specific T cells[59,60]. Thus, the relationship observed in our study between in vitro SEB responses and inflammation could suggest a broad impairment of antigen-specific CD4 T cells to several pathogens. In line with this, studies on T cell immunity in HIV controllers have alluded to the possibility that robust HIV-specific T cell responses are induced in an environment of low inflammation and immune activation[61,62], though a direct relationship remains to be shown. Our findings highlighting defects in antigen-induced T cell activation most likely do not reflect the complete view of how inflammation impairs T cell function. In healthy settings, pro-inflammatory cytokines and type I IFN responses of innate immune cells are critical to T cell responses but excessive inflammation limits antigen presentation[63] and can drive T cell exhaustion or senescence[64]. In the setting of aging, it was shown that CD4 T cells that are activated in an inflammatory milieu are less responsive to type I interferon, as seen by lower levels of CD69+ and IL-2+ cells as well as lower STAT5 phosphorylation[65]. Thus, the relationship between inflammation and reduced vaccine T cell responses in people with HIV could be further exacerbated by aging. Overall, the complex network of pro-inflammatory cytokines such as IL-6, IL8 or TNF and their pleiotropic sources underlines the necessity to study the various immune cells from which these cytokines could originate, such as neutrophils, monocytes or macrophages for a more granular understanding of how inflammation results in aberrant T cell responses.

Our observed relationships between the reduced antigen-specific CD4 T cell responses, altered T cell subset composition and chronic inflammation raise the question whether these could have converged consequences on antibody responses in people with HIV. In contrast to TT, we observed a stronger link between inflammation and MV-specific T cell responses, which may relate to differences in immunogenicity, the time interval between vaccination and the assessment of T cell responses, or the durability of various vaccines. Longitudinal studies assessing the long-term durability of both antibody and T cell responses, especially in the context of antigen-specific T helper subsets such as T follicular helper cells and regulatory T cells, while considering characteristics such as antibody avidity, will help gain a deeper understanding of how inflammation and immune activation influence both cellular and humoral vaccine-induced immunity.

Our assessment of targeting inflammation was only in vitro and is a limitation of our study. Whether targeting inflammation can indeed improve antigen-specific T cell responses in people with HIV will require in vivo studies. It is evident that the outcome of such an intervention could be majorly influenced by factors such as drug bioavailability and duration of treatment, but these aspects cannot be fully mimicked in vitro. Our work established a proof-of-concept for the beneficial effect of tocilizumab and rosuvastatin on memory T cell responses in vitro. Given the well-known challenges of inferring in vivo mechanisms from in vitro interventions, further investigation on the

benefits of TCZ or ROS is best addressed with in vivo studies using drug regimens that were already proven to have anti-inflammatory effects in people with HIV[46,47].

Taken together, our study sheds light on how chronic inflammation—as defined by aberrantly elevated plasma markers—and an altered subsets composition of the T cell compartment could together deteriorate antigen-specific CD4 T cell responses in ART-treated HIV infection and highlights mechanisms that may be relevant for the maintenance of vaccine-induced immunity in other conditions driving chronic inflammation such as aging, auto-immune diseases or cancer.

## Methods
### Experimental design
ART-treated people with HIV and HIV-uninfected healthy adults were recruited at the Universitätsklinikum Erlangen based on prior immunity established in childhood for tetanus toxoid (documented vaccinations) and measles virus (documented vaccinations or infection). The 67 participants had prior immunity to both TT and MV, with 13 of 33 (39%) among the HIV-infected group and 11 of 34 (32%) among the uninfected group having recovered from measles virus infection in childhood. In accordance with national guidelines recommending TT booster administrations, the study participants of both groups had received a median of 5 TT boosters with the latest doses at a median of 6 years (uninfected group) and 5 years (HIV-infected group) from this study. Routine T cell count data of the HIV-infected group were extracted from the clinical registry at the Department of Medicine 3 of the Universitätsklinikum Erlangen. Self-reported absence of HIV infection was confirmed in the uninfected group by a combined measurement of plasma HIV-1 antibodies, HIV-2 antibodies and p24 antigen using the ARCHITECT HIV Ag/Ab combo assay (Abbott) as recommended by the manufacturer. The main objective of the study that analyzed blood samples of the study participant was to assess the quality of antigen-specific T cell responses to TT and MV as well as the relationship between recall T cell responses and inflammation in ART-treated people with HIV in comparison to HIV-uninfected persons.

### Blood samples
Whole blood freshly collected in citrate tubes (Sarstedt) was immediately processed by centrifugation for separation of plasma frozen at −80 °C for subsequent assays. Peripheral blood mononuclear cells (PBMC) were isolated by density centrifugation using Ficoll-Paque Plus (GE Healthcare) and LeucoSep centrifuge tubes (Grenier Bio-One) prior to cryopreservation in liquid nitrogen with heat-inactivated fetal calf serum supplemented with 10% dimethyl sulfoxide (DMSO, Sigma-Aldrich).

### Soluble markers of inflammation and immune activation
Enzyme-linked immunosorbent assay was used to measure plasma levels of sCD14 (R&D Systems), I-FABP (R&D Systems), and sCD163 (R&D Systems); according to the manufacturer's instructions. Plasma concentrations of IL-4, IL-6, IL-8, IL-10, IL-12p70, IFNγ, MIP1β and TNF were measured by Bio-plex magnetic bead-based multiplex assays according to the manufacturer's instructions (Bio-Rad Laboratories Inc). Plasma concentrations of LBP, CRP and CXCL-13 were measured by Human Magnetic Luminex Assay according to the manufacturer's instructions (R&D Systems).

### Measurement of T cell activation and exhaustion
Cryopreserved PBMCs were thawed and washed in R10 media (RPMI1640 supplemented with 10% fetal bovine serum, 2 mM L-glutamine and 100 U/ml penicillin/streptomycin). PBMC were stained with CD3-Cy7APC (BD, clone SP34-2); HLADR-TRPE (Invitrogen, clone TU36); CD8-BV570 (Biolegend, clone RPA-T8), CD4-CY55PE (Invitrogen, clone S3.5), PD1-BV421 (Biolegend, clone EH12.2H7), CD14-PE (BD, clone M5E2), CD19-AF700 (Biolegend, clone HIB19), CD56-FITC

(BD, clone NCAM16.2). For determination as CD57 expression, PBMC were stained with CD57-BV605 (clone QA17A04), CD3-Cy7APC (BD, clone SP34-2); CD8 Pacific Blue (BD, clone RPA-T8) and CD4-CY55PE (Invitrogen, clone S3.5). Prior to antibody staining, all samples were stained with LIVE/DEAD Fixable Aqua Dead Cell Stain (Thermofisher). Stained cells were subsequently acquired on an Attune NxT flow cytometer (Thermofisher) and analyzed with FlowJo v10 (FlowJo LLC).

### Phenotyping of naïve and memory T cell subsets
Cryopreserved PBMCs were thawed and washed in R10 media (RPMI1640 supplemented with 10% fetal bovine serum, 2 mM L-glutamine and 100 U/ml penicillin/streptomycin). PBMC were stained with LIVE/DEAD Fixable Aqua Dead Cell Stain (Thermofisher) then with the following antibodies: CXCR3-BV421 (Biolegend, clone G025H7), CD27-BV711 (Biolegend, clone O323), CD57-FITC (Biolegend, clone HNK-1), CD8-PerCPCy55 (Biolegend, clone RPA-T8), CCR10-PE (Biolegend, clone 6588-5), CD4-PEAF700 (Invitrogen, clone S3.5), CCR7-PECy7 (Biolegend, clone G043H7), CD28-APC (Biolegend, clone 28.2), CD3-AF700 (Biolegend, clone UCHT1), CD45RA-APCCy7 (Biolegend, clone HI100). Stained cells were subsequently acquired on an Attune NxT flow cytometer (Thermofisher) and analyzed with FlowJo v10 (FlowJo LLC). The gating strategy presented in Supplementary Fig. 12 was based on published panels[66,67].

### Measles virus- and tetanus toxin-recall cytokine responses
Cryopreserved PBMCs were thawed in R10 media (RPMI1640 supplemented with 10% fetal bovine serum, 2 mM L-glutamine and 100 U/ml penicillin/streptomycin) supplemented with 20 μg/ml DNase (Sigma-Aldrich), washed in R10 media and rested for 6 h at 37 °C. Cells were aliquoted into wells of a 96-well round bottom plate at total of $1 \times 10^6$ cells per well in R10 medium containing αCD28/αCD49d (BD; 1μg/ml each). For antigenic stimulation, TT protein (Tetanol, GSK−0.5 I.E) or a MV 20-mer overlapping peptide pool corresponding to the measles virus haemagglutinin and fusion proteins (EMC, final concentration 2.5 μg/ml) was added per well. As positive control, PBMCs were stimulated with SEB (Sigma, 0.25 μg/ml final concentration). As negative controls, PBMCs were either unstimulated (TT control) or supplemented with DMSO (0.25% final concentration, MV peptide pool control). Following the addition of all stimuli, cells were incubated at 37 °C 5% $CO_2$ either overnight (for measurement of intracellular cytokine responses) or for 3 days (for Ki67 responses). Antigen-specific T cells have been shown to produce various cytokines, including IFNγ, IL-2 and TNF, up to 30 h into in vitro culture[68]. In our study, we opted for overnight rather than 6 h stimulation as a preliminary test showed marginally better CD69 and IFNγ detection after overnight stimulation. Following overnight culture, cells were incubated for an additional 4 h with 10 μg/ml brefeldin A (Biolegend), 4 μM Monensin (Biolegend) and anti-human CD107a-BV421 (Biolegend, clone H4A3). Next, cells were stained with LIVE/DEAD Fixable Aqua Dead Cell Stain (Thermofisher) prior to surface stain with CD3-Cy7APC (BD, clone SP34-2), CD4-CY55PE (Invitrogen, clone S3.5), CD8-CY55-PERCP (Biolegend, RPA-T8) and CD69-FITC (Biolegend, clone FN50). After fixation with 2% PFA and permeabilization with 0.5% saponin, cells were stained with IFNγ-Cy7PE (Biolegend, clone B27), IL-2-PE (Biolegend, clone MQ1-17H2) and TNF-APC (BD, clone Mab11). For measurement of Ki67 production after 3 days of stimulation, LIVE/DEAD Fixable Aqua stained cells were stained with CD3-PE (BD, clone SK7), fixed with 2% PFA, permeabilized with 0.5% saponin and stained with ki67-AF647 (BD, clone B56). Stained cells were acquired on an Attune NxT flow cytometer (Thermofisher) and analyzed with FlowJo v10 (FlowJo LLC). The gating of stimuli and controls for CD69, IFNγ, IL-2, TNF and CD107a is presented in Supplementary Fig. 13. The presented data are cumulative results of experiments performed over 8 batches, with each batch including participants from both groups. For the analysis of cytokine-positive CD69[+] T cells, cytokine production

was normalized to the frequency of CD69 for each sample prior to subtraction of the background.

## Staining and cell sort of antigen-specific T cells

Cryopreserved PBMC were thawed as described above and rested overnight at 37 °C prior to stimulation for detection of antigen-specific T cells. Briefly, $2 \times 10^6$ cells per well were seeded in R10 medium containing αCD28/αCD49d (BD; 1 μg/ml each) with either SEB (0.25 μg/ml), TT protein (Tetanol, GSK−0.5 I.E), a MV 20-mer overlapping peptide pool corresponding to the measles virus hemagglutinin and fusion proteins (EMC, 2.5 μg/ml) or left unstimulated. After 24 h incubation at 37 °C in the presence of 2 nM monensin and CD154-PE (BD, clone TRAP), cells were stained with LIVE/DEAD Fixable Aqua Dead Cell Stain (Thermofisher) prior to the addition of the following antibodies: CD3-AF700 (BD, clone UCHT1), CD4-APCF750 (Biolegend, clone SK3), CD8-BV421 (Biolegend, clone RPA-T8), CD45RO-BV650 (Biolegend, clone UCHL1) and CD69-FITC (Biolegend, clone FN50). For each stimulation condition, stained cells were sorted an Astrios cell sorter (Beckman Coulter), as shown in the gating strategy in Supplementary Fig. 6. Ten experiment rounds of stimulation and cell sort were performed, with each batch including participants from both groups. Median yields of CD154+ and CD154-memory CD4 T cells were 15,500 and 340,000 cells, respectively (Supplementary Table 5). Following sort of CD4 T cell subsets, cells were resuspended in RNAzol RT (Molecular Research Center) and stored at −80 °C prior to RNA extraction.

## In vitro stimulation with ROS, TCZ, CHIR-99021 and SB145286

Cryopreserved PBMC of 15 ART-treated HIV people from the Erlangen HIV cohort were thawed as described above and rested overnight at 37 °C prior to stimulation with rosuvastatin (Sigma) and tocilizumab (Roche). After 6 h, αCD28/αCD49d (BD; 1 μg/ml each) and SEB (sigma) were added for an additional 15 h. Final concentrations of ROS, TCZ and SEB were 16 μM, 80 μg/ml and 0.2 μg/ml, respectively. Next, 10 μg/ml brefeldin A (Biolegend) and 4 μM Monensin (Biolegend) were added to cells that were incubated for an additional 4 h 37 °C, resulting in a 19 h stimulation with SEB. Cells were then stained with eBioscience Fixable Viability Dye eFluor 780 (Thermofisher) prior to surface stain with CD3-AF700 (Biolegend, clone UCHT1), CD4-FITC (Biolegend, clone RPA-T4), CD8-APC (Biolegend, clone RPA-T8), PD1-BV421 (Biolegend, clone EH12.2H7) and CD45RO-BV650 (Biolegend, clone UCHL-1). After fixation with 2% PFA and permeabilization with 0.5% saponin, cells were stained with CD69-BV510 (Biolegend, clone FN50) and IFNγ-Cy7PE (Biolegend, clone B27). Stained cells were acquired on an AttuneNxt (Thermofisher) and analyzed with FlowJo v10 (FlowJo LLC). For the analysis of responses of T cell subsets, delta values were calculated by subtracting responses of the SEB condition from SEB+ROS or SEB+TCZ for each participant. The percent change was calculated for each participant by dividing delta values by the respective SEB responses and then multiplying by 100. For in vitro stimulation with the GSK3 inhibitors CHIR-99021 and SB415286, cryopreserved PBMC of ART-treated HIV people were thawed as described above, rested and pre-incubated with CHIR-99021 (Sigma), SB415286 (Abcam) or TCZ (Roche) for 2 h, matching the CHIR-99021, SB415286 pre-incubation time that has been described by others to enhance T cell reactivity[69,70]. After the pre-incubation period, αCD28/αCD49d (BD; 1 μg/ml each) and SEB (sigma) were added for overnight incubation at final concentrations of 10 μM or 2.5 μM (CHIR-99021, SB415286), 80 μg/ml (TCZ) and 0.2 μg/ml (SEB). Next, 10 μg/ml brefeldin A (Biolegend) and 4 μM Monensin (Biolegend) was added to cells that were incubated for an additional 4 h at 37 °C. Cells were then stained with eBioscience Fixable Viability Dye eFluor 780 (Thermofisher) prior to surface stain with CD3-AF700 (Biolegend, clone UCHT1), CD4-FITC (Biolegend, clone RPA-T4), CD8-PerCPCy55 (Biolegend, clone RPA-T8), PD1-BV421 (Biolegend, clone EH12.2H7) and CD45RO-BV650 (Biolegend, clone UCHL-1). After fixation with 2% PFA and permeabilization with 0.5%

saponin, cells were stained with CD69-Cy7PE (Biolegend, clone FN50) and IFNγ-APC (Biolegend, clone 4S.B3). Stained cells were acquired on an AttuneNxt (Thermofisher) and analyzed with FlowJo v10 (FlowJo LLC).

## RNA extractions, libraries preparations, mRNA sequencing

RNA was extracted from sorted cells lysed in RNAzol RT (Molecular Research Center), according to the manufacturer's instructions. Briefly, 0.4 volume of sterile water was added to each lysate to allow aqueous and organic phase separation. Total RNA was then extracted from the aqueous phase by isopropanol precipitation. RNA pellets were washed with 70% ethanol and resuspended in sterile water. Yields of extracted RNA were measured using the Qubit RNA HS Assay Kit (Thermofisher) and stored at −80 °C until the preparation of mRNA libraries. Messenger RNA libraries were constructed simultaneously for each subset using the NEBNext Ultra RNA library preparation kit (New England Biolabs). Polyadenylated transcripts were purified with oligo-dT magnetic beads, fragmented, reverse transcribed using random hexamers and incorporated into barcoded cDNA libraries. Libraries were validated by microelectrophoresis on a tapestation 4200 (Agilent), quantified with Kapa Library Quantification Kits (Roche), pooled equimolar and clustered on an Illumina S2 flow cell. Clustered flow cells were sequenced in $2 \times 75$ base paired-end runs on a NovaSeq 6000 (Illumina) during a single run that included all samples.

## Bioinformatics analysis of mRNA sequencing

Raw reads were trimmed for low-quality reads and miscalled nucleotides using Trimmomatic v. 0.36. Reads that passed the trimming step were aligned to the human genome (Ensembl GRCh38 release 88) using the STAR aligner (version 2.5.3a). Aligned reads were counted using HTSeq (version 0.9.1). Next, read count files were merged and genes with mean of <1 read per sample were excluded. The expression and differential expression values were generated using DESeq2 in R[71]. For differential comparisons between groups, an A versus B model with no additional covariates was used. Adjusted $P < 0.05$ and absolute log2fold change of 0.5 was used for significance. PCA, Signature and Over Representation Analysis were performed using the Searchlight pipeline, as described previously[72], where enriched gene sets were determined using the String 11.5 database and a hypergeometric test with Benjamini−Hochberg (BH) correction ($P$-adjust < 0.05). For the gene set enrichment analysis (GSEA), cluster profiler was used[73], specifying a minSize = 3 and maxSize = 800. All other parameters were left to default. Integrative analysis combining our various assays made use of dimension reduction as done previously[34] and was performed in several steps: First, the expression of each assay variable was scaled by feature (across samples). For the gene expression, data was scaled (z-score) by gene (across samples), and subset condition to include only the relevant genes from the GSEA. These enabled estimation of the relative distance between features for any given participant. Then, the mean values combining variable for a given assay outcome (plasma, cell, recall response, Hallmark CD154+ or Hallmark CD154- pathways) were calculated for each individual sample. Finally, the average values (i.e., aggregated distances for each participant) were correlated against each other using a Spearman correlation analysis.

## HIV gag copies in sorted CD4 T cell subsets

DNA was extracted extractions from cells sorted after stimulation with MV or TT using the DNA pellets of the RNAzol RT (Molecular Research Center) extraction according to the manufacturer's instructions. Simultaneous quantification of HIV and human *albumin* (Hu-*ALB*) copies was performed on an ABI7500 Real-Time PCR system with the Luna® Universal Probe qPCR Master Mix (New England Biolabs) and the following primers and probes at final concentrations of 400 nM and 200 nM, respectively for HIV; and 200 nM and 100 nM, respectively for Hu-*ALB*: HIV-*Gag*-F: GGTGCGAGAGCGTCAGTATTAAG; HIV-*Gag*-R: AGCTCCCTG

CTTGCCCATA; HIV-*Gag*-P: Fam-AAAATTCGGTTAAGGCCAGGGGGAAA GAA-Tamra; Hu-*ALB*-F: GTGAACAGGCGACCATGCT; Hu-*Alb*-R: GCATG GAAGGTGAATGTTTCAG; Hu-*Alb*-P: Vic-TCAGCTCTGGAAGTCGATGAA ACATACGTTC-Tamra[74]. Samples were run for 40 cycles along with plasmid standards for absolute quantification of *Gag* and *Albumin* copy numbers. HIV copy numbers were then calculated by comparison to a standard curve, and copies were reported as HIV copies/$10^5$ cells after adjustment based on *ALB* copies in individual samples.

## RT-qPCR of FOXP3, GATA3, SOCS1 and SOCS3

RNA was extracted from PBMC stored in RNAzol RT (Molecular Research Center) as described above, and RNA expression levels of *FOXP3*, *GATA3*, *SOCS1* and *SOCS3* were determined using the Luna Universal One-Step RT-qPCR kit (New England Biolabs) according to the manufacturer's instructions with the following primer sets: *FOXP3*-forward: TCCCA GAGTTCCTCCACAAC, *FOXP3*-reverse: ATTGAGTGTCCGCTGCTTCT[75], *GATA*-3-forward: GCCCCTCATTAAGCCCAAG, *GATA*-3-reverse: TTGTGG TGGTCTGACAGTTCG[76], *SOCS1*-forward: ACCTTCCTGGTGCGCGAC, *SOCS1*-reverse: AGGCCATCTTCACGCTAAGG, *SOCS3*-forward: CCCCCA GAAGAGCCTATTACATCT, *SOCS3*-reverse: GCTGGGTGACTTTCTCAT AGGAG[77], *β-actin*-forward: GCATGGGTCAGAAGGATTCCT, *β-actin*-reverse: TCGTCCCAGTTGGTGACGAT[78]. Specificity and coverage of all known transcript variants were confirmed with PRIMER-BLAST for these primer pairs. Samples were run for 40 cycles on a 7500 real-time PCR system (Applied Biosystems), and fold changes in *FOXP3*, *GATA3*, *SOCS1* and *SOCS3* relative expression were determined using β-actin as reference and calculated using the 2-ΔCt method.

## Human cytomegalovirus (HCMV) serology

HCMV seropositivity was determined be measurement of plasma anti-glycoprotein B (gB) antibodies by ELISA. High binding 96-well Lumi-trac plates were coated overnight with 1.25 µg/well HCMV gB (Sino-Biological). Plates were washed 3 times with PBS-0.05% Tween20 and blocked for 2 h with 5% skimmed milk. Following 3 wash steps with PBS-0.05% Tween20, plasma samples and standards diluted in PBS-0.05% Tween20 and 2% skimmed milk were incubated in the plate for 2 h. Plates were then washed 3 times with PBS-0.05% Tween20 prior to the addition of the secondary antibody Goat-anti-human IgG-HRP (Jackson Immunoresearch). Following 1 h of incubation, plates were washed 3 times with PBS-0.05% Tween20 and then twice with PBS prior to the addition of ECL solution and measurement of relative light units per second (RLU/s) with an Orion microplate luminometer (Berthold Detection Systems GmbH).

## Statistical analysis

Statistical analyses are detailed in the corresponding figure legends. Statistical analyses of univariate comparisons between groups were performed by two-tailed Mann–Whitney test and univariate paired analysis was performed by two-tailed Wilcoxon test using GraphPad Prism. Multivariate analysis of plasma cytokines, cell phenotyping and in vitro recall responses was performed in R by two-tailed Wilcoxon rank sum test with Benjamini–Hochberg correction. Correlations between the various assay measures was perform by Spearman correlation analysis in R. For any given assay, all measurements were taken from distinct samples.

## Study approval

The study protocol was approved by the ethics committees of the Universitätsklinikum Erlangen (235_18B) and carried out in compliance with institutional guidelines. All participants gave written, informed consent in accordance with the Declaration of Helsinki.

## Inclusion and ethics statement

All collaborators of this study have fulfilled the criteria for authorship required by Nature Portfolio journals and have been included as authors, as their participation was essential for the design and implementation of the study.

## Reporting summary

Further information on research design is available in the Nature Portfolio Reporting Summary linked to this article.

## Data availability

This study did not generate new unique reagents. Whole transcriptome sequencing data generated in this study have been deposited in the Gene Expression Omnibus (GEO) database under the series accession number GSE273967. Source data are provided with this paper.

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

## Acknowledgements

This study was supported by the Interdisciplinary Center for Clinical Research (IZKF) at the University Hospital of the University of Erlangen-Nuremberg (Junior project J69) to K.N.M. Additional support was obtained by the Deutsche Forschungsgemeinschaft (DFG) through the research training group RTG 2504 (project number 401821119). We thank Kirsten Fraedrich and Norbert Donhauser for their assistance with the collection of blood samples.

## Author contributions

Conceptualization: K.N.M. Recruitment of study participants: E.H., T.H. and K.N.M. Methodology: M.K., S.K., P.K., A.R., F.L., K.K. and K.N.M. Investigation: M.K., J.J.C., S.F. and K.N.M. Resources: D.C.D. and K.Ü. Visualization: J.J.C. and K.N.M. Funding acquisition: K.N.M. Writing—original draft preparation: K.N.M. Writing—review & editing: All authors. Supervision: K.N.M.

## Funding

## Competing interests

The authors declare no competing interests.
