## [Transparent Peer Review file · Nature Communications]

Chronic inflammation degrades CD4 T cell immunity to prior vaccines in treated HIV infection

Corresponding Author: Professor Krystelle Nganou-Makamdop

Version 0:

Reviewer comments:

Reviewer #1

(Remarks to the Author)

The article by KieBling et al, describes a comprehensive immune analysis combining assessment of frequency, function and transcriptional signatures of measles and tetanus specific T cells and plasma biomarkers in a relatively small cohort of HIV-infected ART treated adults.

The results shed light on impairments within the recall responses of vaccine-specific CD4+ T cells during chronic, ART suppressed HIV infection and show that in vitro culture with drugs targeting inflammation could lead to a reversal of impaired function (i.e. IFN γ production) in the cells pointing to a potential therapeutic intervention approach.

My comments regarding the interpretation and methodology for the study are as follows:

Introduction:

The authors curiously did not address response to influenza vaccination and also only a minor reference to CD8 T cell immunity following vaccination. It would be good to introduce or discuss their study objectives and data in context of these points as well as these are very well reported in the literature.

Results:

Line 98: The assay description should be moved to the methods section.

Line 209: "signatures 2, 4, and 6 mirrored..." Does this mean they are the opposite? Are the genes in each signature unique or are the genes in signature 1 and 2 the same? This point should be clarified.

All of figure 4 could be supplemental as it does not provide any results, rather it is some level of QC for the sorting/RNaseq experiment. The output does not seem to be related to figure 5 and pathway analysis so what is the purpose of the signatures?

Line 279: Fig. 5C+D should be Fig. 5C and D

Methods:

Regarding the measurement of measles virus and tetanus toxin-recall cytokine responses:

I am unclear of the timing used for the intracellular cytokine secretion assay. Overnight plus 4 hours is 20 hours? Standard protocol for ICS assay is 6-12 hours (Lovelace and Maecker, Methods Mol Biol 2011 PMID: 21116983) The authors should reference the protocol they used and justify why it was used over the standard protocol. This detail may be important especially in context of the extremely low frequencies of cytokine+ cells detected (Figure 1B, Supplementary Figure S2). The authors could present the data as cytokine+ CD69+ cells if the markers were included in the same flow cytometry experiment to provide a more specific assessment of Ag-specific responses, especially for CD8+ T cells as shown (<https://doi.org/10.4049/jimmunol.1900856>)

The authors should show the gating and staining examples with controls to support the results presentation (figure 1).

Line 439-441: Why was protein used for TT and peptides used for MV?

Regarding the flow cytometry staining and sort of measles virus- and tetanus toxin-specific T cells

How many cells were sorted from each compartment? Were the compartments equal in cell number, or RNA concentration. A mention of how samples were normalized across the 10 batches of experiments for RNaseq analysis is required. Were the frequencies of CD154+ cells different in the infected and uninfected participants? Does this readout agree with the findings in figure 1?

Line 519: Is the integrative analysis an established method? I am not clear on the methodology/statistics applied here in relation to the result/interpretation.

Discussion

Can the authors comment on HIV-specific CD4 T cells and the impact of immune activation and inflammation?

Also the CD4 compartment is made up of many specialized subsets, including T regulatory and T follicular helper cells, as

well as Th1/Th2/Th17... Could the authors comment on the subsets that could be contributing to the ag-specific cells detected? Or mention as a limitation that the CD4 compartment was not fully characterized.

Were antibody titers to the vaccine antigens measured? This would provide another measure of recall response in the participants.

Line 382: What is the definition of chronic inflammation here? This statement could be toned down as this is a cross-sectional study.

Reviewer #2

(Remarks to the Author)

In this manuscript, Kieβling et al report on the impact of HIV infection on recall vaccine-induced cellular immunity. They studied parameters related to inflammation and immune activation in HIV-uninfected and antiretroviral (ART)-treated HIV-infected people and how these associate with recall T cell responses specific for vaccine antigens (measles virus (MV) and tetanus toxoid (TT)). They report (1) lower recall responses of antigen-specific CD4 T cells correlating with high plasma cytokines levels and T cell hyperactivation in ART-treated HIV-infected people, and (2) transcriptomic alterations of the IFN α and IFN γ response pathways in these antigen-specific CD4 T cells.

This work is well performed and relevant to our understanding of impaired vaccine responsiveness in HIV-infected people. The study is sound and interesting, although it may be considered too preliminary in some aspects. The authors need to provide more information and a deeper analysis of the factors that may affect recall vaccine responsiveness in ART-treated HIV-infected people.

Major comments:

- The authors do not provide any information on the distribution of naïve and memory T cell subsets in their subjects. Could this parameter also associate with the authors' observations on the quantity or quality of MV and TT-specific CD4 T cells?
- The authors need to show or exclude if CMV seropositivity (which affects inflammation and immune activation) play a role in the differences they observe between control and HIV-infected people groups. Does age also affect immune profiles in control or HIV-infected people groups?
- The authors show that levels of several cytokines associates with low recall T cell responsiveness. Is this purely associative, or could there be an underlying biological process at play? For instance, could one or several cytokines in particular affect T cell responsiveness?
- Could the observations made by the authors be actually related to metabolic alterations of antigen specific T cells in ART-treated HIV-infected people. For instance, do metabolic regulators (e.g. inhibitors of GSK3) help restoring the responsiveness of recall T cell responses in vitro?
- There is a rather obvious difference between T cell responsiveness to TT and MV antigens (i.e. MV specific responses are more affected in HIV subjects). Could the authors provide hints as to why there are such differences?

Minor comments:

Figure 1 A&B: Is there explanation why SEB stimulated responses are different between HU and HIV subjects with regards to CD69 but not cytokine secretion? This should be discussed in the manuscript.

Figure 2: the authors classify CD57 as a marker of activation. However, it is more a marker associated with T cell differentiation. To some extent, the expression of HLA-DR and PD-1 can also vary according to naïve and memory T cell differentiation. The authors should take more these aspects into consideration.

Figure 3: how do the authors explain that negative associations between antigen stimulated T cell responses and inflammation makers are more pronounced for the super antigen SEB (compared to MV and TT)?

Figure 4B: is there an explanation behind the difference in cd4 gene expression between CD154+ and CD154- cells?

Version 1:

Reviewer comments:

Reviewer #1

(Remarks to the Author)

The authors have responded acceptably to all previous comments.

Reviewer #2

(Remarks to the Author)

The authors' responses to my comments and changes to the manuscript are satisfactory.

We are very grateful to the Reviewers for their thoughtful comments and insightful criticisms. We have addressed all the concerns with further experiments, analyses, textual changes and we have added the following elements to our revised manuscript: 1 Figure, 5 Supplementary Figures, 1 Supplementary Table, 1 Supplementary Data file. As a result, several sections of the text have been adapted and are highlighted in the revised manuscript. In addition, headings of results and methods sections have been reduced to comply with the Journal resubmission checklist.

Point by point response to reviewers' comments:

Reviewer #1: *The article by KieBling et al, describes a comprehensive immune analysis combining assessment of frequency, function and transcriptional signatures of measles and tetanus specific T cells and plasma biomarkers in a relatively small cohort of HIV-infected ART treated adults. The results shed light on impairments within the recall responses of vaccine-specific CD4+ T cells during chronic, ART suppressed HIV infection and show that in vitro culture with drugs targeting inflammation could lead to a reversal of impaired function (i.e. IFN γ production) in the cells pointing to a potential therapeutic intervention approach.*

We thank the reviewer for these positive comments and for the time taken to carefully review our work.

My comments regarding the interpretation and methodology for the study are as follows:

Introduction:

1- The authors curiously did not address response to influenza vaccination and also only a minor reference to CD8 T cell immunity following vaccination. It would be good to introduce or discuss their study objectives and data in context of these points as well as these are very well reported in the literature.

We thank the reviewer for this suggestion. Our initial manuscript only included 2 references on influenza vaccination in people with HIV and we have expanded this in the revised manuscript to cover various influenza vaccine studies. Given that antiretroviral therapy has an effect on vaccine immunogenicity, we did not include vaccine studies that included mixed cohorts of treated and untreated people with HIV without stratifying immunogenicity based on ART-status (e.g. PMID 18003811). In our revised manuscript, the additional citations reported that (i) among persons who developed seroprotective influenza antibody levels, lower CD154 expression of CD4 T cells (PMID 28017428) and lower PBMC IL-2 responses (PMID 21752440) were observed in people with HIV compared to controls, (ii) the CD4:CD8 T cell ratio associated with a higher magnitude of post-vaccination antibody levels (PMID 21349364) and with higher PBMC IFN γ responses after influenza vaccination (PMID 23623859), (iii) the CD4 T cell recovery did not fully explain failure to induce protective antibody levels (PMID 20616698) - Lines 51 through 67 of the revised manuscript.

While the frequencies of CD8 T cells is often assessed, particularly as the CD4:CD8 T cell ration, CD8 T cell cytokine responses are rarely measured in those studies that have compared immunity induced by licensed vaccines in ART-treated people with HIV and controls. Therefore, we have added reported CD8 T cell data after vaccinia virus vaccination (Line 50 of the revised manuscript) as well as a sentence highlighting this knowledge gap (Line 53 of the revised manuscript).

Results:

2- Line 98: The assay description should be moved to the methods section.

This text has been moved to the method section of the revised manuscript (Line 443).

3- Line 209: "signatures 2, 4, and 6 mirrored..." Does this mean they are the opposite? Are the genes in each signature unique or are the genes in signature 1 and 2 the same? This point should be clarified. The term mirrored was indeed used to describe opposite expression profiles. Each gene only occurs in one signature and a given gene cannot exhibit two different profiles for the same comparison. Thus, each signature contains unique genes. This has been clarified in the revised manuscript (Line 230)

4- All of figure 4 could be Supplementary as it does not provide any results, rather it is some level of QC for the sorting/RNAseq experiment. The output does not seem to be related to figure 5 and pathway analysis so what is the purpose of the signatures?

We agree that the panels A, B and C are important quality control of our cell sort and RNAseq. We have moved these panels to the Supplementary Figure 8 and shorten the description. Panels D and E of the original figure 4, are however not QC analysis. Rather, these demonstrate that the HIV-infected and uninfected groups differ in their global gene expression, especially following stimulation with TT or MV antigens. This a relevant finding to our study objective that aimed at comparing antigen-specific T cell responses between both groups. The original figure 4E highlighted DEGs that are unique to either HIV-infected or uninfected group but, as pointed out by the reviewer, this was not linked to a pathway analysis. To provide clarification on what these unique DEGs are and the associated biological functions, we have added the lists of unique DEGs in a new Supplementary Data 1 and we have performed a pathway analysis on these unique DEGs. The pathway analysis revealed only a few significant pathways – a caveat of the highly stringent DEG filter as explained in the initial manuscript (Line 254 of the revised manuscript). Nonetheless, we could observe that pathways with $P_{adj} \leq 0.05$ in the HIV uninfected group included production and regulation of type I IFN, regulation of leucocyte activation and mononuclear cell differentiation; but these pathways were not found in the HIV infected group. These analysis presented in Figure 5C of the revised manuscript and described at line 238, indeed support our message that antigen-stimulated CD4 T cells of HIV-infected and uninfected persons exhibit distinct transcriptional profiles.

5- Line 279: Fig. 5C+D should be Fig. 5C and D

This has been corrected – now Figure 6C and D - in the revised manuscript (Line 299).

Methods:

6- Regarding the measurement of measles virus and tetanus toxin-recall cytokine responses: I am unclear of the timing used for the intracellular cytokine secretion assay. Overnight plus 4 hours is 20 hours? Standard protocol for ICS assay is 6-12 hours (Lovelace and Maecker, *Methods Mol Biol* 2011 PMID: 21116983) The authors should reference the protocol they used and justify why it was used over the standard protocol. This detail may be important especially in context of the extremely low frequencies of cytokine+ cells detected (Figure 1B, Supplementary Figure S2).

We thank the reviewer for raising this point and to allow us to clarify our experiment rationale. Among various published work, the incubation time with antigens prior to intracellular staining vary greatly. While 6 hours is commonly used, some reports go as much as 24 hours. Chattopadhyay et al has for example reported that their assay – that aimed at identifying antigen-specific CD4 T cells by CD154 staining and is compatible with intracellular cytokine staining – could be used ‘for stimulation times of at least 24h’ (PMID 16186817), with antigen-specific CD4 T cells maintaining steady cytokine production for up to 30 hours. Moreover, Meier et al reported that a 16 hours stimulation did not lower the frequency of CD154+ cells or cytokine production (PMID 18785645). In view of these various reports, we had opted to test which time course would be better in our study and had performed a comparison between 6 hours and overnight (18 hours) incubation prior to assays with our study samples. Upon SEB and MV stimulation, our test did not show a decrease in the frequencies of CD69 positive T cells, but rather a minor increase as shown in the Reviewer Figure 1 below. IFN γ responses to SEB were also higher after overnight stimulation and the frequencies of IFN γ + T cells after MV stimulation was slightly higher in most participants. As such, we decided to proceed with the overnight stimulation. To clarify this, we have added a statement in methods section of the revised manuscript (Line 505).

Reviewer Figure 1

7-The authors could present the data as cytokine+ CD69+ cells if the markers were included in the same flow cytometry experiment to provide a more specific assessment of Ag-specific responses, especially for CD8+ T cells as shown (<https://doi.org/10.4049/jimmunol.1900856>)

We thank the reviewer for this suggestion and for pointing us to the study by Rinaldi et al. We have reanalyzed our data to assess the cytokine+ CD69+ T cells. As stated by others (PMID 17406442), subtraction of the background is necessary to guarantee that specific rather than non-specific responses are measured. Without a background subtraction, frequencies may also appear to be larger. It is unclear if Rinaldi et al included a background subtraction in their analysis as this is not described. In the course of our analysis, it appeared that a simple and direct subtraction of the background is not appropriate because even though small (i.e. in numbers), a few cytokine positive cells in the negative control may make up for a proportionally large fraction of the CD69+ cells. This is exemplified with the Reviewer Figure 2 below, where 4% of IFN γ + cells are seen in the CD69+ gate of the negative control sample compared to 2.7% for SEB stimulation of the same donor. In our opinion, the best way to handle this is to normalize for the CD69 expression prior to the background subtraction. We have added a statement to our methods section to clarify this point (Line 522). In terms of differences between HIV-infected and uninfected groups, our analysis of cytokine+ CD69+ T cells matches responses of total T cell fractions as presented in our initial manuscript. Because CD69 does not guarantee specificity (PMID 23788442) but a presentation of the cytokine production by CD69+ T cells is nonetheless informative, we have included a figure on cytokine+ CD69+ T cells as Supplementary Figure 3 (Line 125).

Reviewer Figure 2

8- The authors should show the gating and staining examples with controls to support the results presentation (figure 1).

We have added the gating of stimuli and controls for CD69, IFN γ , IL-2, TNF and CD107a as Supplementary Figure 13 (Line 519 of the revised manuscript).

9- Line 439-441: Why was protein used for TT and peptides used for MV?

In vitro T cell responses to measles virus are typically performed by co-culture with infected cells or by stimulation with peptide pools as no protein compositions are readily available. For tetanus toxoid, in vitro T cell responses are generally performed with readily available proteins. An interesting aspect of protein stimulation is that it can help assess defects in antigen processing. We initially included MV lysates as additional condition but the MV lysates had an overall poor reactivity and were not further included. In our study, we finally opted for the MV peptide pool and the readily available TT protein because our aim was not to compare MV to TT, but rather to examine them side by side in our comparisons between groups. While some studies reported higher CD8 T cell responses after peptide stimulation compared to whole protein stimulation but no differences for CD4 T cell responses (PMIDs 11470284, 18785645), our aim was also not to compare a protein stimulation to a peptide stimulation, which arguably cannot simply be done by comparing a set of overlapping peptides with the correspondent full protein as the sequence overlap may influence the number of epitopes.

10- Regarding the flow cytometry staining and sort of measles virus- and tetanus toxin-specific T cells How many cells were sorted from each compartment? Were the compartments equal in cell number, or RNA concentration. A mention of how samples were normalized across the 10 batches of experiments for RNAseq analysis is required.

As we were aware that a minimal cell input needed to be met to ensure reliable sequencing data, we kept a good record of cell sort yields. A median of 15,500 CD154⁺ and 340,000 CD154⁻ T cells were sorted. Importantly, all samples were well above the minimum of 500 cells that was established to be sufficient for the library prep protocol. We have added the individual cell sort yields for each subset as a Supplementary file (Supplementary Table 5). We also measured the RNA concentrations after RNA extractions, which very nicely aligned with the cell input as shown in the Reviewer Figure 3. RNA concentrations were only unmeasurable in the low cell samples because of the high quantity input needed to measure concentrations. Nonetheless, good libraries could be confirmed.

Expectedly, higher cell numbers could be sorted for CD154⁻ than for CD154⁺ fractions for any given sample. There was no difference in the sort yield between TT and MV, and HIV and HU groups had a similar distribution as shown in the figure below. It may be added that as difference in RNA input between any two samples is common, the equimolar pooling that is a standard and key step of the library prep protocol that normalizes for such potential differences in input. Thus, the cell sort and RNA yields would not play a role in our sequencing data. As for the 10 batches of experiments, in order to avoid that some samples would be standing for a while (even if one ice) during the cell sort, we had opted to have several small rounds of sorts with few samples rather than big rounds with many samples. This batch approach was for the cell sorting only. All samples were sequenced together and thus normalization between sequencing batches was not needed. To clarify this, we have added statements in the method section regarding the cell sort yields, equimolar pooling and simultaneous sequencing (Lines 539; 583-591).

Reviewer Figure 3

10- Were the frequencies of CD154⁺ cells different in the infected and uninfected participants? Does this readout agree with the findings in figure 1?

For most sorting rounds, the FACS recording during sorting experiments regrettably did not record sufficient events to allow a reliable analysis of the CD154⁺ frequencies. While the yield of CD154⁺ cells shows an enrichment the HIV group among samples with the lowest yields as shown in the figure above, this is not enough to formally prove that frequencies of CD154⁺ were lower in the infected participants. Although we are unable to answer this question within our study, it is worth mentioning that similar to our study, others have reported lower frequencies of CD154⁺ CD4 T cells upon stimulation with antigens (e.g. PMID 28017428, 10444266, 11372024).

11- Line 519: Is the integrative analysis an established method? I am not clear on the methodology/statistics applied here in relation to the result/interpretation.

We thank the reviewer for raising this point and giving us the opportunity to provide clarifications. Dimension reduction is a critical part of integrating multi-omics data (PMID 26969681) as it allows aggregation of a large number of variables into fewer ones, thereby facilitating the graphical presentation of results. As described by Meng et al, this procedure also 'enables all the variables to have equal contribution to the total inertia (sum of squares of all elements) of a data set' (PMID 26969681), thus making it possible to directly compare the outcomes of different assay irrespective of differences in dynamic ranges that are inherent to the different techniques (e.g. flow cytometry vs ELISA vs gene expression). In our study, this the principle that we used to aggregate over 40 dimensions into 5 dimensions presented in Figure 7A of the revised manuscript. Using a similar approach as in prior work (e.g. PMID: 34237254), opting for an arc diagram representation as opposed to a heatmap or chord diagram. In essence, grouping of multiple features was done after multidimensional scaling

across samples/observations, preserving the relative distance between observations; i.e. all features to be combine were scaled across features for each individual participant in order to calculate the relative distance between features for any given participant. Once grouped into the 5 final dimensions, we performed Spearman correlations to assess the relationship between any two reduced dimensions across donors. We have adapted this section in the revised manuscript for clarity (Lines 606-614 of the revised manuscript).

Discussion

12- Can the authors comment on HIV-specific CD4 T cells and the impact of immune activation and inflammation?

We thank the reviewer for raising this point. Published studies that have assessed HIV-specific T cell responses of HIV controllers (elite controllers and/or viremic controllers) and progressors have provided some compelling evidence for the relationship between HIV-specific CD4 T cells and immune activation and inflammation. As highlighted by many studies, HIV controllers - in particular elite controllers - generally have lower levels of T cell activation (e.g. PMIDs 26544698, 29403500, 31308541, 22240463), lower T cell expression of inhibitory receptors (e.g. PMIDs 29403500) as well as distinct plasma cytokine profiles compared to progressors (e.g. PMIDs 30814287, 22240463, 28053103, 36877848). Based on concomitant higher HIV-specific CD4 or CD8 T cell responses of controllers, some of these studies suggested that robust HIV-specific T-cell responses are induced in an environment of low inflammation and immune activation (PMIDs 30814287, 22240463), though a direct relationship remains to be shown. Because the increased expression of immune checkpoints in people with HIV is linked to prolonged T cell activation and numerous studies have shown that cytokine responses of HIV-specific CD4 T cells are improved upon blockade of immune checkpoint inhibitors (reviewed in PMID 38130714), a possible direct and deleterious relationship between chronic immune activation and HIV-specific T cell responses seems plausible but remains to be formally characterized. To address these points, we have added a statement to the discussion section (Line 387 of the revised manuscript).

13- Also the CD4 compartment is made up of many specialized subsets, including T regulatory and T follicular helper cells, as well as Th1/Th2/Th17... Could the authors comment on the subsets that could be contributing to the ag-specific cells detected? Or mention as a limitation that the CD4 compartment was not fully characterized.

We thank the reviewer for this suggestion. We agree that further compartmentalization of antigen-specific T cells into Th subsets may further help understand how inflammation and immune activation may not simply reduce but possible skew T helper responses towards diverse T helper functions. As such, we have added a statement in the discussion to acknowledge this limitation (Lines 404-414 of the revised manuscript).

14- Were antibody titers to the vaccine antigens measured? This would provide another measure of recall response in the participants.

We have measured plasma total IgG against MV and TT and find no significant difference between the groups. While this is very intriguing, antibody concentrations are only a small fraction of what should be considered to assess the quality of humoral responses. For example, it is known that antibody levels wane over a much longer period of time compared to T responses, and thus the time span as well as the slope of decline are important values to consider. In addition, neutralization capacity, affinity and avidity are all measures that are necessary to fully understand how antibody responses may be altered in disease settings. Because these aspects are well beyond the scope of our study that focused on T cell responses, we have refrained from adding a figure on antibody concentrations (Reviewer Figure 4) to the revised manuscript as this would be a very shallow comparison that does not add much to the manuscript. Instead, we have chosen to highlight this interesting aspect raised by the reviewer in the discussion of our revised manuscript (Lines 404-414).

Reviewer Figure 4

15- Line 382: What is the definition of chronic inflammation here? This statement could be toned down -as this is a cross-sectional study.

The persistence of several biomarkers of inflammation, coagulation and innate immune cell activation in people with HIV is often described as chronic inflammation. We have adapted the sentence for clarity and to tone down the conclusion as follow: Taken together, our study sheds light on how chronic inflammation - as defined by aberrantly elevated soluble markers - and an altered subsets composition of the T cell compartment could together deteriorate antigen-specific CD4 T cell responses in ART-treated HIV infection (Line 426 of the revised manuscript).

Reviewer #2 (Remarks to the Author):

In this manuscript, Kießling et al report on the impact of HIV infection on recall vaccine-induced cellular immunity. They studied parameters related to inflammation and immune activation in HIV-uninfected and antiretroviral (ART)-treated HIV-infected people and how these associate with recall T cell responses specific for vaccine antigens (measles virus (MV) and tetanus toxoid (TT)). They report (1) lower recall responses of antigen-specific CD4 T cells correlating with high plasma cytokines levels and T cell hyperactivation in ART-treated HIV-infected people, and (2) transcriptomic alterations of the IFN α and IFN γ response pathways in these antigen-specific CD4 T cells. This work is well performed and relevant to our understanding of impaired vaccine responsiveness in HIV-infected people. The study is sound and interesting, although it may be considered too preliminary in some aspects. The authors need to provide more information and a deeper analysis of the factors that may affect recall vaccine responsiveness in ART-treated HIV-infected people.

We thank the reviewer for these positive comments and for the time taken to carefully review our work.

Major comments:

1- The authors do not provide any information on the distribution of naïve and memory T cell subsets in their subjects. Could this parameter also associate with the authors' observations on the quantity or quality of MV and TT-specific CD4 T cells?

We thank the reviewer for this suggestion. We have performed a more detailed analysis of the T cell compartment to present frequencies of naïve, effector, central memory and effector memory subsets. For the CD4 T cells, people with HIV had lower frequencies of central memory and higher frequencies of both stem cell like memory and effector memory, which was especially driven by an enrichment of intermediate effector memory T cells (Figure 4A of the revised manuscript). For the CD8 T cells, the HIV group had lower frequencies of naïve but higher frequencies of total effector and effector memory subsets that were particularly driven by an enrichment of terminal effector and effector memory cells (Figure 4B of the revised manuscript). Thus, there was an overall enrichment of more differentiated T cell subsets in people with HIV. Furthermore, the frequencies of the cell subsets that were higher in

the HIV group were positively to each other and negatively correlated with both the frequencies of depleted subsets and the T cell recall responses (Figure 4C of the revised manuscript). Incorporation of these T cell subsets frequencies in our integrative analysis resulted in the same outcome as presented in our initial manuscript, where T cell subsets enriched in people with HIV also negatively associated with transcriptional signatures of TT or MV stimulated T cells (Figure 7A of the revised manuscript). Altogether, a detailed characterization of T cell subsets allowed us to establish that next to chronic inflammation and immune activation, altered distribution of naïve and memory T cell subsets in people with HIV also associated with lower antigen-specific T cell responses (Lines 182-195 of the revised manuscript).

2- The authors need to show or exclude if CMV seropositivity (which affects inflammation and immune activation) play a role in the differences they observe between control and HIV-infected people groups. Does age also affect immune profiles in control or HIV-infected people groups?

We thank the reviewer for raising this important point. As often reported for people with HIV, a high proportion of our HIV group, namely 30 of 33 (i.e. 91%) is CMV seropositive. In contrast 18 of 34 (53%) of the uninfected groups are CMV seropositive, aligning with the general prevalence in the healthy population in Germany. The very low number of CMV seronegative HIV did not allow us to investigate inflammation measures between HIV-infected and uninfected persons in absence of CMV. However, we were able to compare all measures of inflammation and immune activation among only CMV seropositive persons. This analysis shows that even among only CMV seropositive, several plasma measures (namely CXCL13, IFABP, IFN γ , IL-4, IL-6, IL-8, IL-10, TNF and sCD14) as well as frequencies of CD4 PD-1⁺ and CD4 CD57⁺ remained higher in the HIV-infected CMV seropositive group compared to the HIV-uninfected CMV seropositive group (Figure 2D revised manuscript). We have added a statement covering this aspect in the revised manuscript (Line 160).

We have also analyzed our data in relation to age and found associations between age and only a few plasma measures, namely CRP, IL-6, IL-8 and IFN γ in the HIV-infected group and IL-8 in the HIV-uninfected group as shown in the Reviewer Figure 5. As for the T cell responses, the overwhelming majority of assay measures did not correlate with age. A detailed tabular overview of these comparisons can be found at the end of this document. Although age did not associate with much of our study variables, we would like to emphasize that, in our opinion, our study participants may not be the most adequate to reliably assess the effect of age. With a median of 41-45 years and only 1 participant in each group being 60 years old or above, the demographic in our study is much younger than the many studies that have reported a link between inflammation and ageing, where participants above 60-70 years of age were included (PMID: 26658771). This raise the question whether a negative effect of inflammation on vaccine-induced immunity in people with HIV could be further exacerbated by age. We have added a statement to this effect in the revised version of our manuscript (Line 396).

Reviewer Figure 5

3- The authors show that levels of several cytokines associates with low recall T cell responsiveness. Is this purely associative, or could there be an underlying biological process at play? For instance, could one or several cytokines in particular affect T cell responsiveness? We thank the reviewer for raising this point. We agree that our analysis on the relationship between the plasma cytokines and T cell responses could be deepened. We initially reported several associations between lower T cell responses and various plasma markers of inflammation such as the cytokines IL-4, IL-6, IL-8, IL-10 and TNF, the monocyte and macrophage activation markers sCD14 and sCD163 as well as with CRP. The highly diverse sources of these soluble markers as well as the cellular pathways through which they act opens numerous possible mechanisms through which T cell responses may be altered. For instance, IL-10 could indicate suppressive function of regulatory T cells whereas IL-4 could indicate a Th2 skewing that suppresses Th1 responses via GATA-3. Next to inhibition of TCR signaling (PMID 32303612), CRP has – similar to IL-6 (PMID 12431386) - also been reported to directly inhibit the Th1 differentiation by driving Th2 polarization through higher GATA-3 and IL-4 expression (PMID 25917100). Moreover, several cytokines including IL-6 and IL-4 may jointly inhibit Th1 responses by inducing an upregulation of suppressor of cytokine signaling (SOCS) proteins, with SOCS1 and SOCS3 having been shown to be main inhibitors of IFN γ responses. To test these hypotheses, we have measured FOXP3, GATA3, SOCS1 and SOCS3 mRNA levels in PBMCs of our study participants and observed that while FOXP3 and GATA-3 mRNA levels were not different between HIV and HU groups, the HIV group had higher levels of SOCS1 and SOCS3 mRNA (Figure 3E of the revised manuscript). In chronic untreated HIV infection, higher rRNA levels of SOCS-1 and SOCS-3 was shown to associate with higher PDL-L1 expression on dendritic cells and with lower cytokine production in response to antigen stimulation (PMID 28056393). This raises the possibility that aberrant regulation by SOCS proteins may be a mechanism by which the plasma cytokines influence T cell responses. These aspects are now highlighted in the results section (Line 177) of our revised manuscript.

4- Could the observations made by the authors be actually related to metabolic alterations of antigen specific T cells in ART-treated HIV-infected people. For instance, do metabolic regulators (e.g. inhibitors of GSK3) help restoring the responsiveness of recall T cell responses in vitro? We thank the reviewer for this suggestion. The higher expression of the oxidative phosphorylation pathways in MV-specific CD154+ T cells of HIV-infected people compared to uninfected persons (Figure 6C) could indicate a metabolic switch. Upon TCR-mediated activation, mTOR mediated suppression of OXPHOS and enhanced glycolysis supports proliferation and differentiation of T cells. Because GSK3 interferes with glycogen metabolism, assessing whether GSK3 inhibitors can improve recall T cell responses in vitro is a very interesting idea. Two hours pre-incubation with the GSK3 inhibitors CHIR-99021 and SB415286 that have effective inhibitory concentrations in the range of 3 to 10 μ M had been

reported to increase IL-12 and TNF production by DC, enhance the migratory behavior of T-cells and to downregulate T cell PD-1 expression (PMIDs 20055739, 34975828, 29312284, 26885856). In our initial test with the two inhibitors, we could confirm that these do not affect cell viability during stimulation and that the highest concentration (10 μ M) reduced PD-1 expression on CD4 T cells, though not consistently on CD8 T cells (Reviewer Figure 6). In our assessment of whether CHIR-99021 and SB415286 could improve recall T cell responses, we could observe that treatment with the inhibitors only did not change CD69 expression but increased IFN γ and PD-1 expression of memory CD4 and CD8 T cells (Supplementary Figure 11A of the revised manuscript). However, upon SEB stimulation and despite lowered PD-1 expression by the 10uM concentrations, both inhibitors lowered CD69 expression and IFN γ production of memory CD4 and CD8 T cells, contrasting with the increased IFN γ production by tocilizumab (Supplementary Figure 11B of the revised manuscript). Thus, GSK3 inhibitors did not improve recall T cell responses in vitro (Line 331 of the revised manuscript). Because of the many enzymes that are affected by such inhibitors and the possibility of off-target effects, it can not be entirely excluded that a targeted regulation of cellular metabolic functions could improve recall T cell responses. It seems this may not be straightforward to asses but it is an avenue we consider exploring in future projects.

Reviewer Figure 6

5- There is a rather obvious difference between T cell responsiveness to TT and MV antigens (i.e. MV specific responses are more affected in HIV subjects). Could the authors provide hints as to why there are such differences?

We thank the reviewer for raising this point. A possible explanation could be differences in the decay over time. Durability of vaccine induced T cell responses, which may be influenced by both the vector and the antigen, could possibly results in different decay over time. While it has been reported that TT humoral immunity wanes faster than that to MV (PMID 17989383), such a comparison has to the best of our knowledge not been done for T cell responses. In the invent that differences in decay over time contributed to the lesser effect on TT T cell responses, longitudinal studies addressing the maintenance of T cell responses in people with HIV and controls may help clarify this point. Furthermore, it is possible that vaccine administered more recently could be less affected on short term than those administered long ago. Our adult study participants had received the last TT vaccines 5-6 years prior to the study whereas MV vaccines were given in the childhood only (i.e. in average 40 years prior). Thus, both the variations in decay and recent vaccine administration could have contributed to a smaller effect on TT responses. To address this point, we have added a statement to the revised manuscript (Lines 404-406)

Minor comments:

6- Figure 1 A&B: Is there explanation why SEB stimulated responses are different between HU and HIV subjects with regards to CD69 but not cytokine secretion? This should discussed in the manuscript. Lower T cell CD69 expression after mitogen stimulation in people with HIV+ has been reported by others (e.g. PMID: 9067659; 9056736; 8528739) and may reflect blunted activation signals. Nonetheless, potent stimulation by superantigens such as SEB may be sufficient to achieve maximal

cytokine production. Early studies by Viola and Lanzavecchia have demonstrated that T cells produce IFN γ when a minimum of TCRs are triggered, and this production rapidly reached a plateau when very high numbers of TCRs are triggered as in the case of superantigen stimulations (PMID 8658175). Thus, despite possibly impaired activation signals in T cells of people with HIV, SEB stimulation may be sufficient to induce saturated cytokine production. To address this points, we have added a statement to the revised manuscript (Line 124). With respect to the lower vaccine immunogenicity and recall T cell responses often seen in people with HIV, an encouraging aspect of the high SEB responses in the HIV group is that these elude to the idea that T cell responses of ART-treated people with HIV can be as high as those of uninfected persons, provided that a sufficiently potent activation occurs. (e.g. by an adjuvant).

7- Figure 2: the authors classify CD57 as a marker of activation. However, it is a more a marker associated with T cell differentiation. To some extent, the expression of HLA-DR and PD-1 can also vary according to naïve and memory T cell differentiation. The authors should take more these aspects into consideration.

The reviewer makes a valid point that the expression of these markers is not sufficient to define a subset. We aimed here to highlight the expression of surface markers that have been found to be elevated in T cells of people with HIV, were associated with T cell dysfunction or were linked to immune activation and inflammation (e.g. 16339584, 21917895, 26627102) and thus had included total frequencies of CD57, PD1 and HLA-DR in this analysis. Furthermore, Shive et al reported that with the exception of unaltered PD1 expression on CD8 EM, the CD57 and PD-1 expression of naïve, central memory and effector memory T cells of people with HIV was higher (PMID 26627102). We have added a statement in our revised manuscript (Line 151) to clarify our interest and focus on cell surface markers that are uncharacteristically higher on T cells of people with HIV rather than defining these as subsets.

8- Figure 3: how the authors explain that negative associations between antigen stimulated T cell responses and inflammation makers are more pronounced for the super antigen SEB (compared to MV and TT)?

SEB has been shown to induce very high frequencies of antigen-specific CD4 T cells with as high as 15-30% of stimulated cells expressing CD154 (PMIDs 16186817, 16186818, 17114495). Upon stimulation in the presence of anti-CD28 and anti-CD49d antibodies that provide optimal co-stimulatory signals for the detection of antigen-specific cytokine production by CD4 and CD8 T cells (PMIDs 14580882, 17406442), we likely activated and measured a larger pool of antigen-specific T cells in people with HIV that, similar to TT- and MV-antigen-specific T cells, exhibit defects associated with chronic inflammation and immune activation. We believe measurement of this larger pool of antigen-specific T cells increased the sensitivity of detecting these relationships and have added a statement in the revised manuscript to this effect (Line 381).

9- Figure 4B: is there an explanation behind the difference in cd4 gene expression between CD154+ and CD154- cells?

The regulation of CD4 gene expression during T cell activation is not entirely understood. Some studies have shown that CD4 expression transiently reduced during the first 48 hours of stimulation (10843673). In a study comparing gene expression between resting and activated CD4 T cells, the CD4 expression varied during the course of stimulation and, at 16hours post-stim, CD4 appeared to be among the group of genes (GM2) that showed lower expression in CD4 effector memory compared to naïve CD4 T cells (PMID: 35618845 – suppl information). In absence of studies that have investigated CD4 gene expression in antigen-specific T cells, we can only speculate that CD4 gene expression could be differently regulated between CD154+ and CD154- T cells, particularly during antigenic stimulation. This, however, remains to be clarified.

Correlation between age and study variable in HIV and HU

Variable	HIV		HU	
	Spearman r	P	Spearman r	P
sCD14	0.1591	0.3765	0.1133	0.5237
IFABP	0.2353	0.1874	-0.02054	0.9082
sCD163	0.133	0.4607	0.2462	0.1604
MIP1 β	0.1155	0.522	-0.006322	0.9717
IL-6	0.4779	0.0049	-0.1894	0.2835
IFN γ	0.4278	0.013	0.3706	0.0309
TNF	0.3094	0.0798	0.1263	0.4766
IL-4	0.2891	0.1027	-0.004359	0.9805
IL-8	0.3726	0.0327	0.3598	0.0366
IL-10	0.2898	0.1018	-0.08343	0.639
IL-12	0.1004	0.5783	0.02591	0.8843
LBP	0.2138	0.2322	-0.07494	0.6736
CRP	0.4236	0.014	0.252	0.1504
CXCL13	-0.06994	0.6989	-0.1254	0.4798
CD4_DR ⁺	0.1114	0.5372	0.06841	0.7007
CD4_PD1 ⁺	0.2057	0.2509	0.3264	0.0596
CD8_DR ⁺	0.1825	0.3093	0.3232	0.0622
CD8_PD1 ⁺	0.03082	0.8648	0.6076	0.0001
CD8_CD57 ⁺	0.2704	0.1412	0.3818	0.0449
CD4_CD57 ⁺	0.179	0.3188	-0.1691	0.339
TT_CD4_CD69 ⁺	-0.1194	0.553	-0.04506	0.8131
TT_CD4_IFN γ ⁺	-0.2453	0.2174	-0.1661	0.3804
TT_CD4_IL2 ⁺	-0.1089	0.5886	-0.08155	0.6684
TT_CD4_TNF ⁺	-0.2549	0.1994	0.1465	0.4397
TT_CD8_CD69 ⁺	-0.2047	0.3057	-0.06654	0.7268
TT_CD8_CD107a ⁺	-0.03301	0.8702	-0.3662	0.0465
TT_CD8_IFN γ ⁺	-0.2403	0.2273	-0.1138	0.5493
TT_CD8_IL2 ⁺	-0.04437	0.8261	-0.103	0.5879
TT_CD8_TNF ⁺	-0.3651	0.0611	0.1714	0.3652
TT_CD3_CD69 ⁺	-0.2155	0.2803	-0.1048	0.5814
TT_CD3_IFN γ ⁺	-0.3404	0.0823	-0.1331	0.4831
TT_CD3_IL2 ⁺	-0.2021	0.3122	-0.03547	0.8524
TT_CD3_TNF ⁺	-0.4276	0.0261	0.1635	0.3879
TT_CD3 ⁺ Ki67 ⁺	-0.2966	0.1331	-0.05175	0.786
TT_CD3 ⁻ Ki67 ⁺	-0.5	0.0079	0.01541	0.9356
MV_CD4_CD69 ⁺	-0.03989	0.8434	0.09087	0.633
MV_CD4_IFN γ ⁺	0.3092	0.1166	-0.1038	0.5852
MV_CD4_IL2 ⁺	0.3185	0.1054	-0.004168	0.9826

MV_CD4_TNF ⁺	-0.03293	0.8705	0.1517	0.4236
MV_CD8_CD69 ⁺	0.04321	0.8305	0.2704	0.1484
MV_CD8_CD107a ⁺	-0.146	0.4673	0.2139	0.2564
MV_CD8_IFNg ⁺	0.2683	0.1761	0.0614	0.7472
MV_CD8_IL2 ⁺	0.09319	0.6438	0.156	0.4103
MV_CD8_TNF ⁺	-0.1295	0.5198	0.3481	0.0594
MV_CD3_CD69 ⁺	-0.03519	0.8617	0.2327	0.2159
MV_CD3_IFNg ⁺	0.09666	0.6315	0.06543	0.7312
MV_CD3_IL2 ⁺	0.01942	0.9234	0.05401	0.7768
MV_CD3_TNF ⁺	-0.147	0.4645	0.2232	0.2358
MV_CD3 ⁺ Ki67 ⁺	0.1485	0.4598	-0.384	0.0362
MV_CD3 ⁻ Ki67 ⁺	-0.04939	0.8067	-0.4775	0.0076
SEB_CD4_CD69 ⁺	-0.161	0.4225	0.0418	0.8264
SEB_CD4_IFNg ⁺	0.02173	0.9143	-0.2065	0.2735
SEB_CD4_IL2 ⁺	0.1316	0.5129	-0.2002	0.2888
SEB_CD4_TNF ⁺	-0.3249	0.0982	0.4294	0.0179
SEB_CD8_CD69 ⁺	-0.3608	0.0645	-0.3496	0.0582
SEB_CD8_CD107a ⁺	-0.07774	0.6999	-0.4995	0.0049
SEB_CD8_IFNg ⁺	-0.09535	0.6362	0.2631	0.1601
SEB_CD8_IL2 ⁺	-0.03981	0.8437	-0.4231	0.0198
SEB_CD8_TNF ⁺	-0.3948	0.0416	0.3391	0.0668
SEB_CD3_CD69 ⁺	-0.2556	0.1982	0.005618	0.9765
SEB_CD3_IFNg ⁺	-0.1073	0.5943	-0.08529	0.6541
SEB_CD3_IL2 ⁺	0.08784	0.6631	-0.07415	0.697
SEB_CD3_TNF ⁺	-0.3482	0.0751	0.4501	0.0126
SEB_CD3 ⁺ Ki67 ⁺	-0.1867	0.3511	-0.3031	0.1035
SEB_CD3 ⁻ Ki67 ⁺	-0.2277	0.2533	-0.3512	0.057